# Gray whale (*Eschrichtius robustus*) post-mortem findings from December 2018 through 2021 during the Unusual Mortality Event in the Eastern North Pacific

Stephen Raverty[1]*, Pádraig Duignan[2], Denise Greig[3,4], Jessica L. Huggins[5], Kathy Burek Huntington[6], Michael Garner[7], John Calambokidis[5], Paul Cottrell[8], Kerri Danil[9], Dalin D'Alessandro[10], Deborah Duffield[10], Moe Flannery[4], Frances MD Gulland[11], Barbie Halaska[2], Dyanna M. Lambourn[12], Taylor Lehnhart[8], Jorge Urbán R.[13], Teri Rowles[14], James Rice[15], Kate Savage[16], Kristin Wilkinson[17], Justin Greenman[18], Justin Viezbicke[18], Brendan Cottrell[19], P. Dawn Goley[20], Maggie Martinez[2], Deborah Fauquier[14]

1 Animal Health Center, Abbotsford, BC, Canada, 2 The Marine Mammal Center, Sausalito, CA, United States of America, 3 Ocean Associates, Inc. under contract to Office of Protected Resources, National Marine Fisheries Service, National Oceanic and Atmospheric Administration, Silver Spring, MD, United States of America, 4 California Academy of Sciences, San Francisco, CA, United States of America, 5 Cascadia Research Collective, Olympia, WA, United States of America, 6 Alaska Veterinary Pathology Services, Eagle River, AK, United States of America, 7 Northwest ZooPath, Monroe, WA, United States of America, 8 Fisheries and Oceans Canada, Vancouver, BC, Canada, 9 Southwest Fisheries Science Center, National Marine Fisheries Service, National Oceanic and Atmospheric Administration, La Jolla, CA, United States of America, 10 Portland State University, Department of Biology, Portland, OR, United States of America, 11 Wildlife Health Center, School of Veterinary Medicine, UC Davis, Davis, CA, United States of America, 12 Washington Department of Fish and Wildlife, Marine Mammal Investigations, Lakewood, WA, United States of America, 13 Departamento de Ciencias Marinas y Costeras, Universidad Autónoma de Baja California Sur, La Paz, Mexico, 14 Office of Protected Resources, National Marine Fisheries Service, National Oceanic and Atmospheric Administration, Silver Spring, MD, United States of America, 15 Oregon State University Marine Mammal Institute, Hatfield Marine Science Center, Newport, OR, United States of America, 16 Alaska Biosystems under contract to Alaska Regional Office, National Marine Fisheries Service, National Oceanic and Atmospheric Administration, Juneau, AK, United States of America, 17 Protected Resources Division, West Coast Region, National Marine Fisheries Service, National Oceanic and Atmospheric Administration, Seattle, WA, United States of America, 18 Protected Resources Division, West Coast Region, National Marine Fisheries Service, National Oceanic and Atmospheric Administration, Long Beach, CA, United States of America, 19 Applied Remote Sensing Lab, Department of Geography, McGill University, Montreal, QC, Canada, 20 Department of Biological Sciences, Cal Poly Humboldt, Arcata, CA, United States of America

☯ These authors contributed equally to this work.
¤ Current address: University of British Columbia, Vancouver, BC, Canada
* stephen.raverty@gov.bc.ca

**Data Availability Statement:** Third party data are publicly available from NOAA (https://www.

## Abstract

Beginning in December 2018, increased numbers of gray whale (*Eschrichtius robustus*) strandings were reported along the west coast of Mexico, the United States, and Canada, prompting declaration of a gray whale Unusual Mortality Event (UME) by the United States National Marine Fisheries Service. Although strandings declined in 2020 and 2021 from a peak in 2019, the UME is still ongoing as of fall 2023. Between 17 December 2018 and 31 December 2021, 503 animals stranded along the west coast of North America, with 226 strandings in Mexico, 71 in California, 12 in Oregon, 56 in Washington, 21 in British

fisheries.noaa.gov/national/marine-life-distress/2019-2023-gray-whale-unusual-mortality-event-along-west-coast-and). Additional data is available as Supporting Information.

**Funding:** National Marine Fisheries Service provided support in the form of salaries for authors DF and PC, respectively, but did not have any additional role in the study design, data collection and analysis, decision to publish, or preparation of the manuscript. The specific roles of these authors are articulated in the 'author contributions' section. The funders had no role in study design, data collection and analysis, decision to publish, or preparation of the manuscript.

**Competing interests:** The authors have no competing interests.

Columbia, and 117 in Alaska. These included 187 males, 167 females, and 149 whales of undetermined sex; and 193 adults, 194 subadults, 40 calves, 1 fetus, and 75 whales of undetermined age class. We report on 61 of the 503 carcasses (12%) that had external and internal gross necropsy and/or histopathology data: of these 61 whales, findings that contributed to death were identified in 33 (54%) whales. Sixteen of the 61 (26%) were severely emaciated. Gross lesions of blunt force trauma consistent with vessel strike were identified in 11 of the 61 animals (18%), only two of which were emaciated. Two whales (3%) were entangled at time of death, and one died from entrapment. Signs of killer whale (*Orcinus orca*) interaction were documented in 19 of the 61 animals; five were deemed from recent interactions and three (5%) likely contributed to mortality. A specific cause of death could not be identified in 28 of 61 whales (46%). Additionally, logistical challenges and the advanced state of decomposition of most examined carcasses precluded detection of potential infectious or toxic causes of morbidity or mortality. Up to 2016, the eastern North Pacific population of gray whale population had generally been increasing since the cessation of historic whaling and a prior UME in 1999–2000. However, recent abundance and calf production estimates have declined, a trend that overlaps the current UME. The relative contributions of carrying capacity, environmental change, prey shifts, and infectious, toxic, and other processes to the increased gray whale mortalities have not yet been resolved. Nevertheless, the marked temporal increase in strandings, including findings of malnutrition in some of the whales, along with low calf production, likely represent consequences of complex and dynamic ecological interactions in the ocean impacting the population.

## Introduction

There are two genetically distinct gray whale (*Eschrichtius robustus*) populations: the eastern north Pacific (ENP) and the western north Pacific (WNP) [1]. The ENP whales range seasonally from wintering grounds off Mexico to primary feeding areas in Arctic waters, with the exception of a small subset, the Pacific Coast Feeding Group (PCFG). The latter are often observed feeding in summer and fall from northern California to southeastern Alaska [2]. The endangered WNP population has been observed feeding off Sakhalin Island, Russia [1]. Historically, gray whales were thought to fast during their migration and while resident on the wintering grounds [3]; however, more recent field observations indicate some animals actively forage during the southward and northward migrations [4–7]. Whales from both populations have been observed on the wintering grounds in Mexico and along the ENP migratory corridor, but the extent of mixing is not clear [1, 2, 8–10]. Heavily exploited during the 1800s and 1900s by commercial whaling, the ENP population has made a strong recovery, with abundance estimated in the mid-2010s at approximately 27,000 [11, 12], the largest number documented for this population [13]. Models of gray whale abundance and mortality estimated carrying capacity in recent years approximating 25,000 [14].

Gray whale abundance in the 1850s was probably less than 30,000 [15], although genetic studies have estimated that higher numbers existed through evolutionary time [16, 17]. The 2015/2016 gray whale abundance estimate was 26,960 individuals, which declined to 20,580 in 2019/2020 and further to 16,650 in 2021/2022 [18]. Since 1967, similar decreases in estimated abundance were detected between 1987/1988 and 1992/1993 and between 1997/1998 and 2000/2001, which coincided with the 1999–2001 Unusual Mortality Event (UME) [18].

Transformative changes in the summer Bering Strait feeding grounds [19], with reported local declines in prey abundance and quality [20], may have a direct impact on gray whale bio-energetics, fecundity, health, and homeostasis [21]. Diet varies, by geographic location and whether foraging in shallow mud flats or deeper waters, and consists primarily of benthic and epibenthic invertebrates, including amphipods, crustaceans, mysids, and plankton [20, 21]. Variability in calf production in the ENP gray whale population over a 23-year period was associated with the extent of seasonal ice cover in the Bering Sea in the previous feeding season and thought to reflect the greater spatial and temporal access to benthic feeding provided by reduced ice cover [18, 22, 23]. For the WNP gray whales, a correlation was observed between decreased calf production and a shorter foraging season (i.e., fewer ice-free days) off Sakhalin Island [24]. Decreased body condition was documented for PCFG whales in years following poor upwelling conditions [25]. Periods of low calf production occurred during periods of increased mortality [4, 18, 26], and the most recent estimate of calf production for the ENP population was the lowest of the time series, with 217 calves in 2022 [27].

In addition to the impacts of a changing environment, increased vessel traffic and extensive commercial fisheries along migratory corridors throughout the western seaboard of North America have been associated with vessel strikes and entanglements of gray whales [28, 29]. These traumatic events may be immediately lethal or incapacitate animals due to pain, secondary infection, and the incurred energetic costs of wound repair and resolution [30]. In a review of gray whale entanglements and vessel strikes in the North Pacific between 1924 and 2015, 40% of injuries and mortalities were due to net fisheries, 17% to pot fisheries, 22% were from unknown fisheries entanglements, and 19% were from vessel strike [29]. Other potential threats include acute or chronic noise exposure, physical disturbances, disruption of feeding habitats, infectious diseases, harmful algal blooms, toxic spills, pollutants, predation, subsistence harvest, and scientific research [14].

The earlier 1999/2001, gray whale UME resulted in 283 confirmed strandings in 1999 and 368 in 2000 [31], however, only three whales were necropsied during this event and while the cause of death for most whales was not determined, malnutrition was considered a prime contributor to the UME [31]. Of the three whales necropsied, one animal had encephalitis, another had a heavy intestinal helminth infection, and a third had high levels of domoic acid, a biotoxin, in the gut contents [31, 32]. However, the combination of thin and emaciated stranded animals, migrating whales in poor body condition, reduced calf production, a population suspected to be nearing carrying capacity, and reduced prey densities in these years, likely contributed to this earlier event [31, 33–36].

Beginning in December 2018, increased numbers of gray whale strandings were reported in Mexico (MX) and the continental United States (US), including California (CA), Oregon (OR), and Washington (WA). In response to this, the United States National Marine Fisheries Service (NMFS) declared an UME in 2019 (https://www.fisheries.noaa.gov/national/marine-life-distress/2019-2023-gray-whale-unusual-mortality-event-along-west-coast-and). Designation of the UME facilitated coordination of stranding and necropsy responses in three countries, enhanced communication amongst stranding response teams, and heightened public awareness through increased reporting by the media. Here we present the pathology findings from a subset of the gray whales that stranded along the North American west coast between December 2018 and December 2021.

## Materials and methods

Marine mammal mortalities observed by the public, biologists, enforcement, and indigenous community members were reported to Canadian, Mexican, or US regional marine mammal

stranding response networks, and when feasible, trained personnel were mobilized to examine stranded whales. The teams typically included experienced field biologists and veterinarians who collected basic data on these animals [e.g., NMFS Level A data (NOAA Form 89–864; OMB Control No.0648-017)] including species, date, sex, age class, stranding location, and signs of human interaction (e.g., entanglement, vessel strike).

Photographs of the carcasses from different perspectives were used to assess the nutritional status, cyamid load, identifying features of individual animals, evidence of fisheries interactions, killer whale attacks, and any other external lesions. Interpretation of vessel strike was based on photographs plus internal and external examination.

Age class was initially determined based on a combination of length from tip of the snout to the fluke notch (calf <7–8 m, yearling 8–9 m, subadult 9–11.1 m for males and 9–11.7 m for females, adult ≥11.1 m for males and ≥ 11.7 m for females). This was subsequently modified depending on date and latitude of stranding, baleen plate length, and barnacle size, as calves and subadults in the lower latitudes were often shorter than the previously reported length ranges [3, 37–41]. Yearling and subadult age classes were combined because. Outside Mexico, 1–2 year-olds could not be distinguished from 2–3 year olds, therefore final age classes were calf/fetus (0–12 months), subadult (12 months-sexual maturity), and adult. The calf cohort includes fetuses from the calving lagoons in MX and a single fetus documented in British Columbia (BC), Canada in 2021 [42].

Defining emaciation for fasting migratory whales is complicated (for example, blubber depth, while easily measured at post mortem examination (PME), is not a sensitive measure of nutritional status in gray whales [31]). Because of a lack of available quantitative tests to assess nutritional condition, and to avoid applying different measures of nutrition in different locations, a protocol was developed during this investigation and standardized across all stranding response networks to ensure consistent interpretation of post-mortem nutritional condition (S1 Appendix). The protocol used external features (e.g. the concavity of the epaxial muscle), internal features (e.g. epicardial and mesenteric fat stores), and microscopic evaluation of full thickness blubber sections (e.g. for evidence of adipocyte atrophy or stromal collapse). As body position or posture, the stage of putrefaction, bloat, scavenging, and post mortem oil seepage from the blubber may alter the gross appearance and thickness of the blubber, microscopic findings of a disproportionate increase in stroma to adipocytes and fat atrophy were interpreted in context of the gross observations to assign the overall nutritional score for an animal. For the more challenging cases, gross and histopathology findings were reviewed by pathologists, biologists and prosectors, to derive the nutritional score. Carcasses were assigned into one of four nutritional scores: Emaciated (1), thin (2), average (3) and fat (4). Furthermore, the carcass condition was defined as alive (1), fresh dead (2), moderate decomposition (3), advanced decomposition (4), mummified (5), or undetermined (6) [43]. Carcass code was assigned at the time of examination or, if the carcass was not examined, at the time of reporting. Where the carcass code was scored as between decomposition states (e.g. 2.5 or 3.5), [40], the post-mortem state was assigned to the more advanced decomposition state for tabulation and analysis.

PMEs were performed as thoroughly and systematically as possible and was contingent on multiple variables including, but not limited to, carcass access (eg. beach cast or floating), state of decomposition, human safety (including restrictions imposed by the Covid19 pandemic in 2020), tides, available daylight, and weather. For fresher floating carcasses or those with external evidence of traumatic injuries, efforts were made to tow the animal ashore to a secure and accessible site. In some instances, examination of floating animals was limited to photographs above and below water levels using submerged waterproof cameras, and sampling of skin and blubber for genetics and histopathology. Criteria for inclusion in this study were a full gross PME and, for most cases, sampling for histopathology. Exclusions were based on incomplete

PMEs or microscopic assessment of sampled tissues, even if they were confirmed, probable, or suspect cases of human or killer whale interaction. As such, the numbers reported herein are a minimum and represent a subset of the full UME response and investigation.

Depending on the orientation of the animal, typically the skin and blubber from the flanks was flensed, reflected, and the blubber was assessed for nutritional status and trauma. Trauma was inferred by evidence of epidermal lacerations or incisions, hemorrhage, edema and necrosis or maceration of skeletal muscle, and skeletal fractures. Whereas entanglement was documented based on presence of foreign material attached to the carcass and determined to have been present at the time of death [44]. The abdominal and thoracic cavities were then incised, the ribs removed, body cavities examined, and appropriate tissue samples harvested. Response level varied by geographic location and in some cases, access to internal organs was limited to "windows" or small incised portals that facilitated access to specific viscera [45].

Killer whale predation as a cause of death was determined based on lesions that have been documented during observed predation events including missing portions of the tongue or the mandibles, semilunar tissue defects (bite wounds), detached of sheets of blubber and skin, tooth impressions ("rake marks"), and fractured or avulsed bones, subcutaneous and muscular contusions, and frank hemorrhage within the thoracic or abdominal cavities and associated tissues [46–48]. Predation was differentiated from post-mortem scavenging by the presence of associated superficial or deep hemorrhage and edema in the tissues and abrupt margins associated with the rake marks and on histology, lack of imbibed water or melanocyte degranulation.

A case definition of killer whale predation was formulated based on the literature to consistently evaluate the gross lesions and assign a level of diagnostic certainty (S2 Appendix). Four levels of certainty were attributed: confirmed, probable, suspect, improbable, and could not be determined.

Pigment deposition along the wound margins, second intention healing, and varying degrees of re-epithelialization were used to discriminate a peracute injury, from a more chronic and resolving process that we considered incidental to mortality. In cases that presented with advanced autolysis, the wound margins were more closely scrutinized and if assessments were inconclusive, were not considered predation events. Other factors associated with the strandings (clinical history), field observations or reports of transient killer whales in the vicinity and near the time of the mortality, and exclusion of vessel strike and other disease processes were included in diagnosis of a predation event.

For histopathologic examination and ancillary testing of selected cases, 10% formalin-fixed and fresh tissue samples were collected, and forwarded to a diagnostic laboratory for analysis, or frozen and retained for follow up investigations. Fixed tissues were trimmed, placed in cassettes, and processed using conventional histologic methodology. Tissues were stained with hematoxylin and eosin (H&E) and when indicated, Tort's Gram stain, Periodic Acid Schiff (PAS) and Perl's stain (for iron) for review by veterinary pathologists. When feasible, baleen, ear plug (wax), stomach contents, full thickness blubber, and ovaries were archived for future studies.

For this review of stranding and post-mortem records, the stranding location, demographic data (e.g., NMFS Level A), and necropsy reports were uploaded to a shared Google drive and a spreadsheet of cases was maintained by responders. A team of veterinary pathologists and field personal experiences with post mortem examination of large cetaceans reviewed each of the 61 cases to assign a cause of death. Where discrepancies occurred, the field personnel involved with the necropsy and tissue sampling were consulted and photographs at the time of necropsy reviewed. The gross and histologic findings were compiled by a research scientist and two veterinary pathologists with summary tables forwarded to regional stranding responders and investigative team members to review for accuracy. Reported stranding locations (in decimal degrees) were mapped using ArcGIS Pro 2.8.0 with Natural Earth base maps (naturalearthdata.com).

## Results

### Stranding response and demographics

Between 17 December 2018 and 31 December 2021, 503 gray whale strandings were reported along the Pacific coast of MX (n = 226, 44.9% of total), the US (n = 256, 50.9%) and Canada (n = 21, 4.2%) (Table 1, Fig 1). Strandings occurred in each month of the year, with peak numbers in April (n = 99, 19.7%) (Table 2). The proportion of strandings by region included 71 in CA (14.1%), 56 in WA (11.1%), 12 in OR (2.4%), 21 in BC (4.2%) and 117 in AK (23.3%) (Table 1). In contrast to BC, WA, OR, CA and MX where dead whales were reported by the public, biologists, fisheries officers, or Indigenous community members, most of the reports for AK were from aerial surveys or opportunistic aerial sightings of floating or beach cast animals. PMEs from 21 of 71 (29.6%) animals that stranded in CA, 1 of 12 (8.3%) animals in OR, 26 of 56 (46.4%) animals in WA, 9 of 21 (42.8%) animals in BC, and 4 of 117 (3.4%) animals stranded in AK are reported here (Table 1). There were no PME data from the 226 animals stranded in MX.

The demographics of all the reported gray whales, as well as the subset discussed in this paper (in parentheses) are described in Tables 3 and 4 by location and year. The only newborn noted outside of Mexico was a fetus reported in BC [42].

Carcass decomposition varied with 10 mummified animals (2.0%), 251 (50.0%) in advanced decomposition, 175 (34.8%) in moderate decomposition, 51 (10.1%) fresh dead, and 3 (0.6%) that were initially observed alive (Table 5).

### Gross findings

Of the 61 selected whales, a cause of death was attributed for 33 cases (54%, Table 6, Figs 1 and 2). This included 16 whales with emaciation as the only post mortem finding, 11 whales with evidence of vessel strike (including two that were also emaciated), three whales with pre-mortem killer whale attack (two probable, one suspect), two entanglement cases, and one entrapment.

**Nutritional condition.** Using the standardized nutritional condition protocol (S1 Appendix), there were sufficient data to assess the nutritional condition of 55 of 61 examined whales (Table 7). Nutritional scores for five animals could not be determined for various reasons including limited photographic documentation, post-mortem bloat, posture when beach cast (e.g. dorsal recumbency), substrate contour and other on-site variables. In five cases, the addition of microscopic evaluation of the blubber changed the nutritional status assigned grossly:

**Table 1. Gray whale strandings by location and year (17 December 2018 to 31 December 2021).** MX = Mexico, CA = California, OR = Oregon, WA = Washington, BC = British Columbia, AK = Alaska. The number in parentheses is the subset of whales for which a full PME and sampling were conducted.

| Location | Number of Strandings per Year (Number of Post-mortem Exams) | | | | |
|---|---|---|---|---|---|
| | **2019** | **2020** | **2021** | **Total** | **% Strandings** |
| MX | 83* | 88 | 55 | 226 | 44.9% |
| CA | 34 (12) | 18 (2) | 19 (7) | 71 (21) | 14.1% |
| OR | 6 (1) | 3 | 3 | 12 (1) | 2.4% |
| WA | 34 (16) | 13 (8) | 9 (2) | 56 (26) | 11.2% |
| BC | 11 (3) | 5 (4) | 5 (2) | 21 (9) | 4.2% |
| AK | 48 (4) | 45 | 24 | 117 (4) | 23.1% |
| **Total** | **216 (36)** | **172 (14)** | **115 (11)** | **503 (61)** | **100%** |

*2 whales from December 2018 in Mexico are included in the 2019 tally

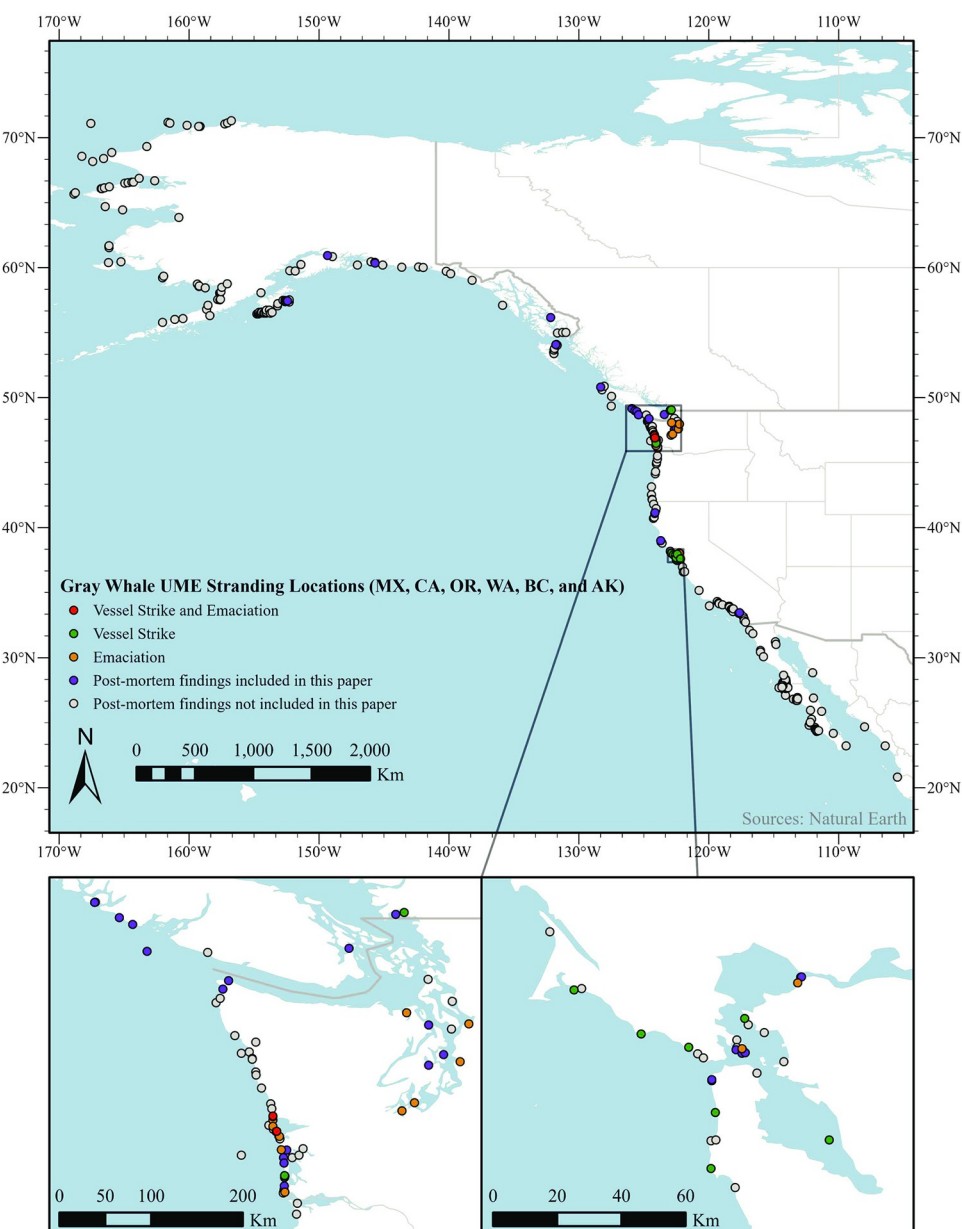

**Fig 1. Map of gray whale strandings from Dec 2018 through 2021.** The map shows the locations of all 503 whales reported, and the vessel strike and emaciation cases among the 61 whales included in this study. Inset maps show area details around Puget Sound and San Francisco Bay.

two cases were changed from average to thin, two cases changed from thin to emaciated, and one case that could not be determined grossly was categorized as thin (Fig 3).

Additional reported gross observations included diffuse salmon pink blubber (12 cases in WA and two in BC), as well as subcutaneous edema in two animals each stranded in WA, CA, and BC and one animal stranded in AK (Fig 4). The reason for the color variation of the blubber was not determined but was observed in the more severely malnourished animals (seven were emaciated and seven were thin). Stomach contents were not consistently present and/or recorded, but wood chips, bark, eel grass, kelp, and some prey were identified in a few animals (Fig 5).

**Table 2. Gray whales stranded by month in Mexico, the United States and Canada (17 December 2018 to 31 December 2021).** MX = Mexico, CA = California, OR = Oregon, WA = Washington, BC = British Columbia, AK = Alaska. The number in the parentheses is the subset of whales examined and sampled.

| Month | MX | CA | OR | WA | BC | AK | Total |
|---|---|---|---|---|---|---|---|
| **Number of Strandings (Number of Post-mortem Exams)** | | | | | | | |
| January | 31 | 1 | 0 | 1 | 0 | 0 | **33** |
| February | 60 | 4 | 1 | 2 (1) | 0 | 0 | **67 (1)** |
| March | 61 | 19 (4) | 1 | 4 (3) | 0 | 0 | **85 (7)** |
| April | 55 | 21 (10) | 4 | 14 (7) | 4 (3) | 1 | **99 (21)** |
| May | 11 | 21 (4) | 2 | 19 (10) | 5 (1) | 13 (2) | **71 (17)** |
| June | 0 | 1 (1) | 3 (1) | 6 (2) | 4 (1) | 32 (1) | **46 (6)** |
| July | 0 | 0 | 0 | 4 (2) | 4 (3) | 36 (1) | **44 (6)** |
| August | 0 | 1 | 1 | 4 (1) | 1 | 14 | **21 (1)** |
| September | 0 | 1 | 0 | 0 | 0 | 15 | **16** |
| October | 0 | 0 | 0 | 1 | 1 (1) | 5 | **7 (1)** |
| November | 0 | 1 (1) | 0 | 0 | 2 | 1 | **4 (1)** |
| December | 8 | 1 (1) | 0 | 1 | 0 | 0 | **10 (1)** |

The primary cause of death was starvation in 16 of 18 (89%) of the emaciated whales, with no apparent pre-existing pathology that would have contributed to suboptimal nutritional condition. The other two emaciated whales had signs of blunt force trauma, which was the primary cause of death. Among the animals with thin and average nutritional condition, there were other processes which may have contributed to or resulted in the death of the animal including blunt force trauma attributed to ship strike (n = 8), suspect propeller wound (n = 1), possible killer whale attack (n = 2), entrapment (n = 1), gear entanglement (n = 1), and historic inactive entanglement (n = 1).

**Vessel strike.** Vessel strike was the cause of death in 11 of 61 cases (Table 7, Fig 2C–2E). Seven vessel strike cases were identified in CA, primarily in the San Francisco Bay area, three in WA and one in BC. Nutritional status varied for these cases: two were emaciated, three were thin, five were average and one was fat. One additional whale had a suspected propeller wound on its flipper, but the contribution of this injury to stranding could not be determined. Another whale had healed propeller scars that likely did not contribute to stranding.

**Gear interactions and entrapment.** One whale was reported entrapped in wharf pilings, and this contributed to its death. Five whales had evidence of active and inactive gear

**Table 3. Gray whales stranded in Mexico, the United States and Canada (17 December 2018 to 31 December 2021) by age class and geographic location from south to north.** MX = Mexico, CA = California, OR = Oregon, WA = Washington, BC = British Columbia, AK = Alaska. The number in the parentheses is the subset of whales included in this study.

| Location | Age Class | | | | |
|---|---|---|---|---|---|
| | Calf/Fetus | Subadult | Adult | Unknown | Totals |
| MX | 22 | 98 | 103 | 3 | 226 |
| CA | 6 (2) | 32 (8) | 27 (11) | 6 | 71 (21) |
| OR | 1 | 9 (1) | 1 | 1 | 12 (1) |
| WA | 3 | 27 (10) | 22 (16) | 4 | 56 (26) |
| BC | 1* (1) | 5 (2) | 9 (6) | 6 | 21 (9) |
| AK | 8 | 23 (2) | 31 (2) | 55 | 117 (4) |
| **Total** | **41 (3)** | **194 (22)** | **193 (35)** | **75** | **503 (61)** |

*Indicates the one fetus, the other 40 whales were calves.

**Table 4. Gray whale strandings by sex and year (17 December 2018 to 31 December 2021).** MX = Mexico, CA = California, OR = Oregon, WA = Washington, BC = British Columbia, AK = Alaska. The number in the parentheses is the subset of whales with a full post mortem examination (PME).

| Location | Number of Strandings per Year (PME)) | | | |
|---|---|---|---|---|
| | **2019** | **2020** | **2021** | **Total** |
| Male | | | | |
| MX | 20 | 34 | 32 | 86 |
| CA | 10 (3) | 7 (1) | 10 (3) | 27 (7) |
| OR | 2 | 3 | 2 | 7 |
| WA | 14 (8) | 5 (3) | 3 (1) | 22 (12) |
| BC | 1 (1) | 4 (3) | 2 (2) | 7 (6) |
| AK | 9 (3) | 20 | 10 | 39 (3) |
| Female | | | | |
| MX | 52 | 25 | 16 | 92 |
| CA | 14 (9) | 9 (1) | 7 (4) | 30 (14) |
| OR | 3 (1) | 0 | 0 | 3 (1) |
| WA | 14 (8) | 4 (4) | 4 (1) | 22 (13) |
| BC | 6 (2) | 0 | 1 | 7 (2) |
| AK | 6 (1) | 6 | 1 | 13 (1) |
| Unknown | | | | |
| MX | 11 | 29 | 7 | 47 |
| CA | 10 | 2 | 2 | 14 |
| OR | 1 | 0 | 1 | 2 |
| WA | 6 | 4 (1) | 2 | 12 (1) |
| BC | 4 | 1 (1) | 2 | 7 (1) |
| AK | 33 | 19 | 13 | 65 |
| **Total** | **216 (35)** | **172 (14)** | **115 (11)** | **503 (61)** |

*2 whales from December 2018 in Mexico are included in the 2019 tally

entanglements. Two stranded with active entanglements that contributed to death (one entanglement that may have resulted in drowning and one with a pectoral flipper amputation and gear present). The proximate cause of death of these animals has not yet been fully resolved but may be associated with exertional or capture type myopathy [49, 50].

In three whales, the entanglements did not lead to death (one whale had a partially healed entanglement scar, and two had healed entanglement scars). These animals presented with

**Table 5. Gray whales stranded in Mexico, the United States and Canada (17 December 2018 to 31 December 2021) by decomposition state and geographic location from south to north.** MX = Mexico, CA = California, OR = Oregon, WA = Washington, BC = British Columbia, AK = Alaska. Decomposition state is recorded at the time of examination, however for whales that were not examined, decomposition state at is recorded at first observation. The number in the parentheses is the subset of whales included in this study.

| Location | First observed alive | Fresh Dead | Moderate Decomposition | Advanced Decomposition | Mummified | Unknown | |
|---|---|---|---|---|---|---|---|
| MX | 1 | 11 | 71 | 137 | 6 | | 226 |
| CA | 1 | 16 (7) | 38 (13) | 16 (1) | | | 71 (21) |
| OR | | 1 | 7 | 3 (1) | | 1 | 12 (1) |
| WA | | 3 (2) | 27 (19) | 24 (5) | 2 | | 56 (26) |
| BC | 1 | 4 (3) | 1 (2) | 7 (4) | | 8 | 21(9) |
| AK | | 16 (2) | 31 (1) | 64 (1) | 2 | 4 | 117 (4) |
| **Total** | **3** | **51 (14)** | **175 (35)** | **251 (12)** | **10** | **13** | **503 (61)** |

**Table 6. Causes of death and location of stranded gray whales (n = 61).** CA = California, OR = Oregon, WA = Washington, BC = British Columbia, AK = Alaska.

| Causes of Death | CA | OR | WA | BC | AK | Total |
|---|---|---|---|---|---|---|
| Emaciation | 2 | 1 | 13 | | | **16** |
| Vessel strike | 7 | | 3 | 1 | | **11** |
| Active entanglements | | | 2 | | | **2** |
| Entrapment | | | 1 | | | **1** |
| Killer whale attack | 1 | | | 1 | 1 | **3** |
| Undetermined | 11 | | 7 | 7 | 3 | **28** |
| **Total** | **21** | **1** | **26** | **9** | **4** | **61** |

superficial linear furrows, often with corrugated to well delineated margins circumferentially involving the axilla or fluke. At PME, the cutaneous defects were chronic with second intention healing and in one animal, there was focal secondary subcutaneous abscessation. Advanced autolysis and position of the carcass at the time of examination may have obscured similar lesions in other whales. While these lesions were not associated with the stranding, they may have impacted the health or nutritional state of the animal.

An adult male had been satellite tagged seven years prior to death and at necropsy, an appropriate granulation tissue response was associated with a fragment of the tag that remained in the blubber. While this whale was in thin body condition, the cause of death could not be determined [51].

**Interactions with killer whales and sharks.** Nineteen cases, eight adults and 11 subadults, had lesions consistent with killer whale interaction characterized by acute lacerations or chronic (healed) scars. The latter were observed on 14 whales predominantly on the tips of the fins, flukes, or more rarely along the rostrum. Five carcasses had more acute active lesions consistent with recent killer whale predation, and the likelihood that predation contributed to mortality was categorized as probable for two cases (eg. Fig 2F and 2G), suspect for one case, and could not be determined in two cases (Fig 2H). Nutritional condition scores for these five whales were variable: one emaciated, two thin, one average and one that could not be determined. For the 14 animals with chronic non-lethal killer whale interaction, 10 were emaciated, and four were thin. Also observed in three gray whales were smaller and more round to oval superficial cutaneous defects circumscribed by regular serrated margins, consistent with agonal or post-mortem shark bites.

## Histologic findings

Although post-mortem decomposition and bacterial overgrowth frequently hampered microscopic assessment of various tissues, especially internal viscera, histologic findings substantiated the gross observations of emaciation and trauma and revealed significant findings in 33 of 61 (54%) examined animals. Microscopic findings were categorized by organ system (Table 8). The most common findings included blubber atrophy (n = 40, Fig 3B), hepatic hemosiderosis (n = 15, Fig 6A), ulcerative or erosive dermatitis (n = 14), muscle atrophy (n = 10, Fig 6B), muscle edema (n = 11), muscle necrosis (n = 8), muscle degeneration (n = 6, Fig 6C), muscle hemorrhage (n = 5), and pulmonary edema (4), aspiration (n = 3), and pneumonia (n = 3). Sporadic and incidental diagnoses included adrenal and myocardial hemorrhage, penile papilloma, myocellular sarcocystosis (Fig 6D), glossitis and lingual mucosal inclusions (Fig 6E and 6F), gastric erosions and enteric cestodiasis (Table 8). Histologic assessment of blubber showed a gradient in atrophy from the deepest to more superficial layers and featured prominent variation in the size and shape of adipocytes, and cytoplasmic condensation (atrophy or lipid depletion) with a disproportionate abundance of stroma (Fig 3).

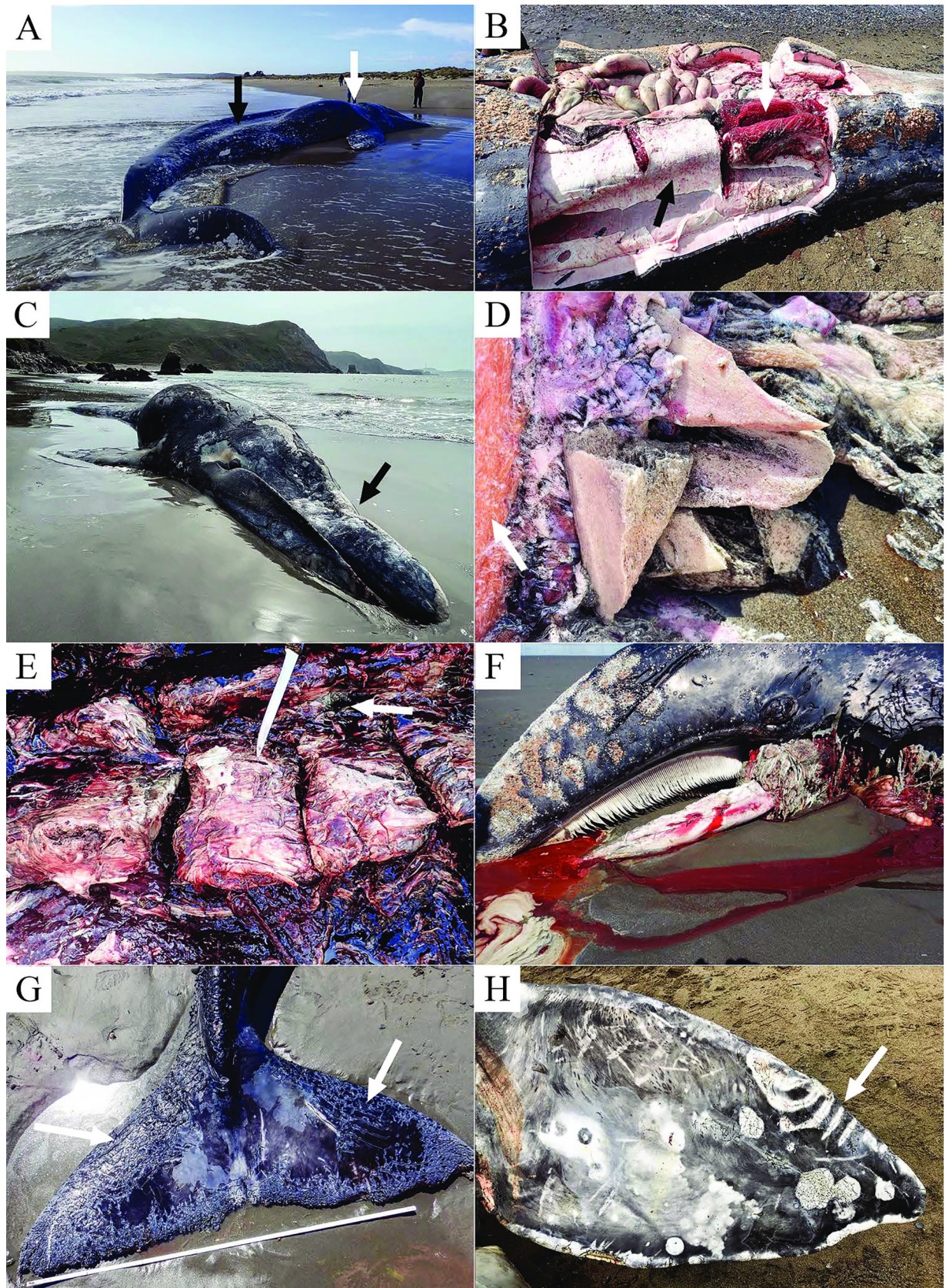

**Fig 2. Examples of gross pathology findings.** (A) Adult female gray whale category 1 emaciation showing loss of nuchal adipose tissue (white arrow) and severe epaxial muscle atrophy (black arrow); (B) Sub-adult male gray whale with blubber flensed from the right side to reveal severely atrophied hypaxial muscle (white arrow) and epaxial muscle (black arrow); (C) Sub-adult female gray whale with concave rostral deformity (arrow) caused by blunt force trauma; (D) Comminuted fracture of the rostral bones of whale C, arrow shows subjacent hypodermal contusion; (E) Adult male gray whale with exposed thoracic vertebrae showing transverse fractures of the dorsal vertebral processes (indicated by knife point) and muscular hemorrhage (arrow); (F) Gray whale calf predated by killer whales with loss of the tongue and soft tissue from the mandibles; (G) Flukes from the same calf showing acute epidermal lacerations consistent with killer whale tooth rakes (arrows); (H) Subadult gray whale pectoral flipper showing killer whale tooth rakes on the leading edge (arrow).

## Discussion

In December of 2018, increased numbers of dead gray whales were first reported stranded in Mexico, with subsequent increases in mortality in 2019 over the northward migration spanning from CA to OR, WA, BC, and ultimately, AK. The resulting UME declaration by the NMFS enhanced the response and pathologic assessment of stranded whales. While emaciation was speculated to have had a role in the previous 1999/2000 gray whale UME, this investigation confirmed its significant role in the 2019/2021 event. Despite standardization achieved through the nutritional condition protocol (S1 Appendix), further work is needed to characterize the nutritional status of dead gray whales. Microscopic evaluation of blubber and other tissues was found to be of value in this investigation and when combined with validated quantitative biomarker measurements should be invaluable with future nutritional health assessments.

The gradation in atrophy of adipose tissue through the blubber layer reported here suggests that greater effort is required to differentiate atrophy associated with physiologic fasting from atrophy of malnutrition. This would better inform decisions on body condition and enable researchers to factor it into the nutritional condition score matrix in the future (S1 Appendix). Because of logistical challenges at the time of PME and varying states of tissue decomposition, not all organs or tissues were systematically examined or collected, so histologic lesions were likely underrepresented.

Interpretation of nutritional status of stranded whales can be further improved by the integration of ecologic and environmental parameters as well as body condition indices from live gray whales; for example, photogrammetry has detected a decrease in body condition in 2018 compared to previous years [25]. In addition, these methods were used to describe reduced body condition in live gray whales in Mexico in 2018 and 2019 compared to 2017 [52]. Both studies coincide with our investigation: 2019 is the year when we detected the greatest number of emaciated whales (n = 14).

As gray whales are usually thin during the northward migration because they have been fasting, it is difficult to assign thin or emaciated body condition as a cause of death; however, in

**Table 7. Nutritional Score of stranded gray whales (n = 61).** CA = California, OR = Oregon, WA = Washington, BC = British Columbia, AK = Alaska. CBD = Could not be determined.

| Location | Nutritional Score | | | | |
|---|---|---|---|---|---|
| | **Emaciation (Poor)** | **Thin (Fair)** | **Average (moderate)** | **Fat (Excellent)** | **CBD** |
| CA | 2 | 9 | 6 | 1 | 3 |
| OR | 1 | 0 | 0 | 0 | 0 |
| WA | 15 | 10 | 0 | 1 | 0 |
| BC | 0 | 5 | 2 | 0 | 2 |
| AK | 0 | 3 | 1 | 0 | 0 |
| **Total** | **18** | **27** | **9** | **2** | **5** |

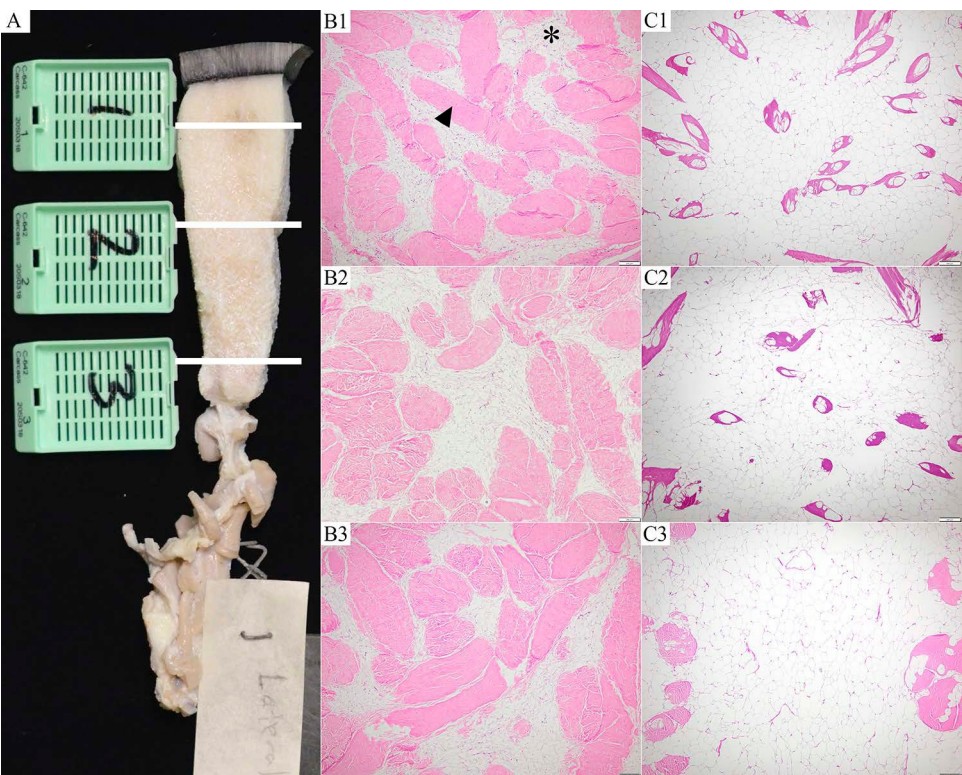

**Fig 3. Blubber sample topography and histologic features.** (A) Gray whale blubber full depth sample including the epidermis and panniculus muscle. White bars indicate sampling locations for the superficial, mid and deep blubber; (B1) Superficial blubber, adult female, showing marked adipose atrophy (star) and relative abundance of connective tissue (arrowhead); (B2) Mid blubber layer from the same whale; (B3) deep blubber layer from the same whale; (C1) Superficial blubber layer, adult female, good body condition; (C2) Mid blubber layer and C3, deep blubber layer from the same well-conditioned whale.

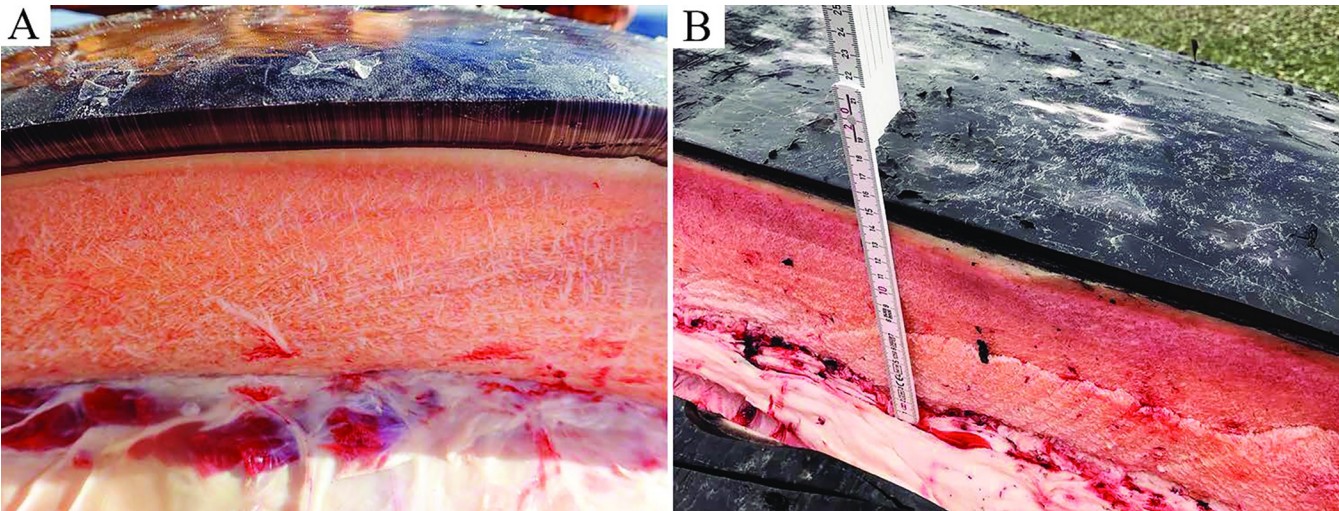

**Fig 4. Gray whale blubber cut section.** (A) Adult female in emaciated condition with pale pink blubber. (B) Adult female gray whale in thin condition stranded in Washington State with diffusely salmon pink blubber.

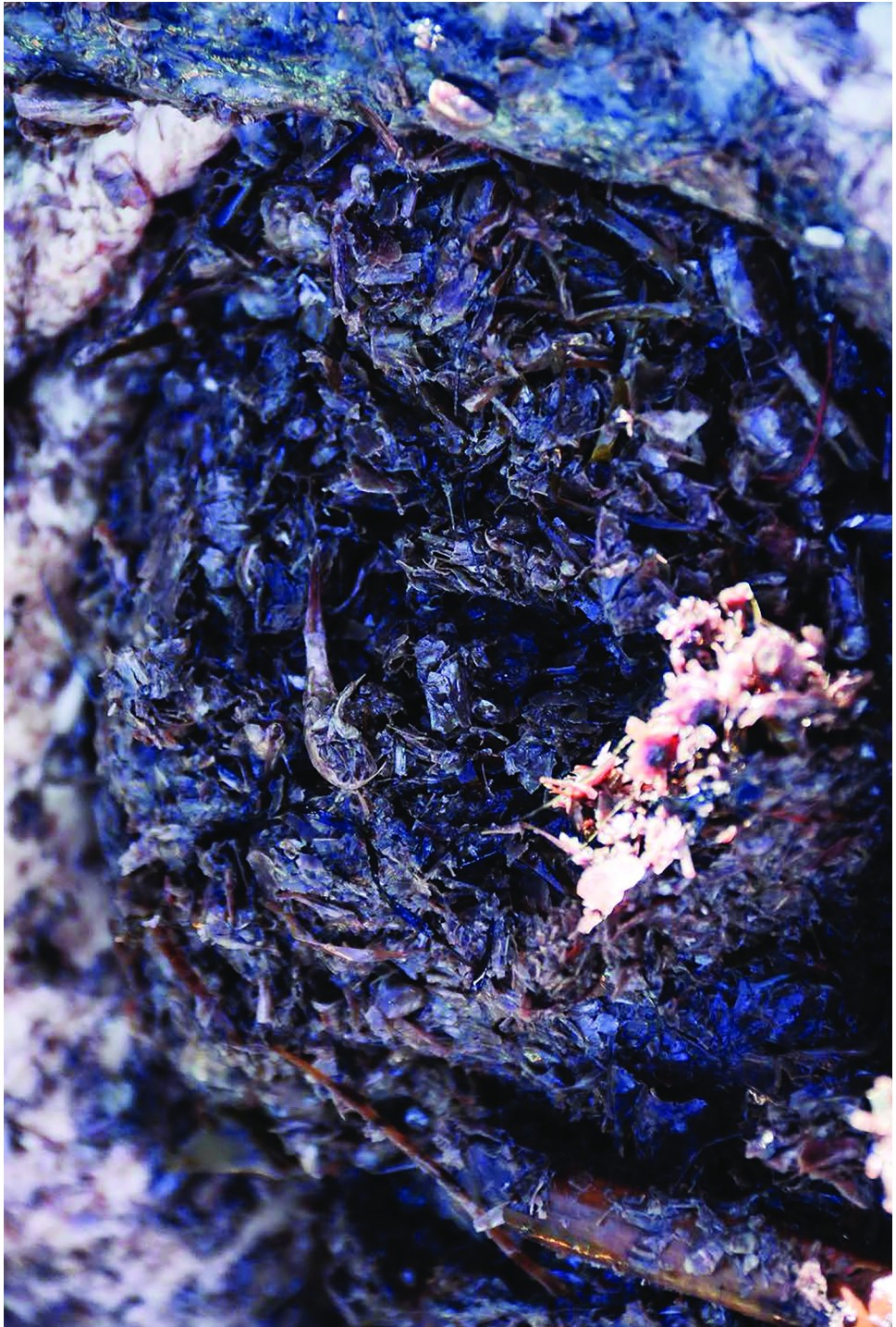

**Fig 5. Adult female gray whale ingesta.** Stomach from an adult female gray whale opened to show content composed predominantly of plant-based fibers and a few amphipod-like crustaceans.

more severely emaciated animals (code 1 nutritional status, Fig 2A and 2B), malnutrition would presumably have contributed to antemortem morbidity, dehydration, loss of buoyancy and eventual live or dead stranding [31]. Ectoparasite (Cyamidae, whale lice) infestations (Fig 7)

**Table 8. Gross and microscopic lesions identified in stranded gray whale tissues from 53 of 61 whales.** The data were generated by reviewing histopathology reports and compiling frequency of morphologic diagnoses. CA = California, OR = Oregon, WA = Washington, BC = British Columbia, AK = Alaska. Multiple morphologic diagnoses in individual tissue sections accounted for a total number of diagnoses greater than the 53 of 61 animals that were evaluated microscopically.

| Morphologic diagnoses | CA | OR | WA | BC | AK | Totals |
|---|---|---|---|---|---|---|
| **Skin and Blubber** | | | | | | |
| Dermatitis | 3 | 0 | 5 | 3 | 1 | **12** |
| Ulcerative or erosive dermatitis | 1 | 0 | 6 | 6 | 2 | **15** |
| Fat necrosis | 0 | 0 | 0 | 0 | 0 | **0** |
| Blubber atrophy | 13 | 0 | 19 | 7 | 1 | **40** |
| **Skeletal Muscle** | | | | | | |
| Muscle atrophy | 7 | 0 | 0 | 3 | 0 | **10** |
| Muscle edema | 6 | 0 | 1 | 4 | 0 | **11** |
| Muscle necrosis | 4 | 0 | 2 | 1 | 1 | **8** |
| Muscle degeneration | 3 | 0 | 0 | 2 | 1 | **6** |
| Muscle hemorrhage | 5 | 0 | 0 | 0 | 0 | **5** |
| Muscle Sarcocysts | 1 | 0 | 0 | 0 | 1 | **2** |
| **Lung** | | | | | | |
| Pulmonary edema | 1 | 0 | 1 | 2 | 0 | **4** |
| Pulmonary aspiration | 1 | 0 | 1 | 1 | 0 | **3** |
| Pneumonia | 1 | 0 | 0 | 1 | 1 | **3** |
| Myocardial hemorrhage | 1 | 0 | 0 | 0 | 0 | **1** |
| **Gastrointestinal and Liver** | | | | | | |
| Gastric erosions | 0 | 0 | 1 | 0 | 0 | **1** |
| Enteric cestodes | 0 | 0 | 1 | 0 | 0 | **1** |
| Tongue cytoplasmic inclusions | 1 | 0 | 0 | 0 | 0 | **1** |
| Liver hemosiderosis | 2 | 0 | 11 | 2 | 0 | **15** |
| Liver atrophy | 0 | 0 | 0 | 1 | 0 | **1** |
| **Miscellaneous** | | | | | | |
| Adrenal hemorrhage | 1 | 0 | 0 | 0 | 0 | **1** |
| Penile papilloma | 0 | 0 | 1 | 0 | 0 | **1** |
| Liver inclusions | 1 | 0 | 0 | 0 | 0 | **1** |

and secondary cutaneous ulcers and erosions were consistently noted in emaciated animals in this case series and the loss of buoyancy in more extreme cases of malnutrition may have resulted in carcasses lost at sea, rather than potentially submerged, refloated, beach cast, and thus available for necropsy [53].

Important sequelae to loss of blubber mass or insufficient caloric consumption in the summer feeding grounds may also include infertility (anestrous), early embryonic loss and fetal resorption, abortion, premature birth, death of dependent calves, live stranding, a reduced ability to refloat and possible death due to asphyxiation or a compartmental-type syndrome [50]. In live stranding situations, animal welfare considerations and consultation with appropriately experienced clinical veterinarians is warranted [54]. In the longer term, malnourished females may experience reduced fecundity, prolonged inter-calving interval, or decreased life longevity, which can lead to low calf recruitment as documented in the recent calf production surveys [27].

In many cases, the physiologic demands associated with prolonged migration, parturition, lactation, and reproduction may account for the decline in nutritional condition in females; however, the increased proportion of emaciated whales documented in WA relative to Mexico, CA, and OR suggests metabolic consequences due to depleted fat stores, prolonged inappetence, possible anorexia and other factors during the northward migrations along the west coast. Unidentified environmental or infectious factors may have also contributed to the

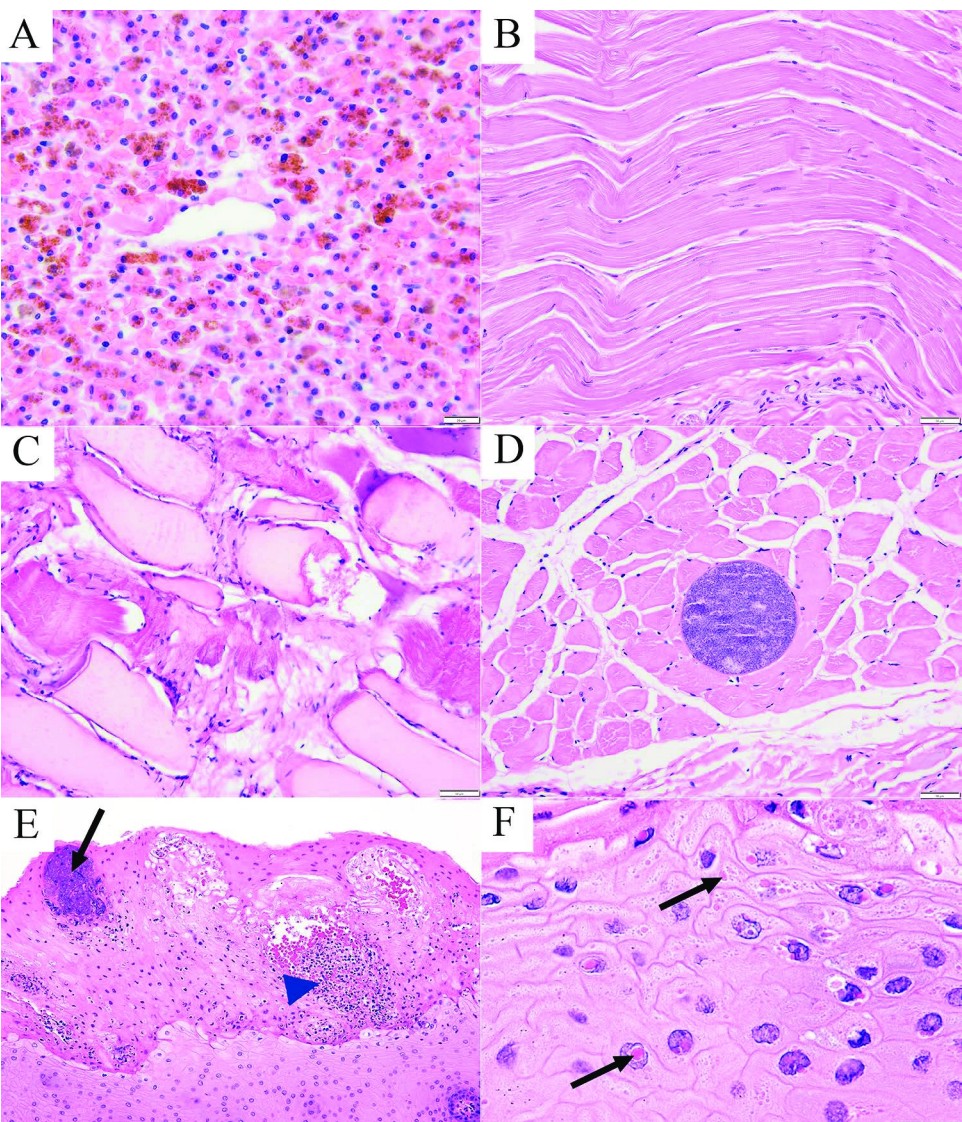

**Fig 6. Representative histopathology of stranded gray whales.** (A) Liver, adult female gray whale, emaciated body condition, marked diffuse hemosiderosis, H&E stain, x400 magnification, bar = 20μm; (B) Skeletal muscle, subadult male gray whale, thin body condition, moderate diffuse myocyte atrophy. H&E stain, x200 magnification, bar = 50μm; (C) Skeletal muscle, adult female gray whale, emaciated, severe diffuse myocyte degeneration. H&E stain, x200 magnification, bar = 50μm; (D) Skeletal muscle, adult male gray whale, thin body condition, moderate diffuse myocyte atrophy and protozoal tissue cyst, *Sarcocystis* spp. presumptive. H&E stain, x200 magnification, bar = 50μm; (E) Tongue, gray whale calf, glossitis with epidermal hyperkeratosis, hydropic change and necrosis of acanthocytes with neutrophilic infiltration (intraepithelial pustules, blue arrow head) and intralesional bacterial colonies (black arrow). H&E stain, x100 magnification, bar = 100μm; (F) Tongue, gray whale calf, glossitis as in E, acanthocytes with intracytoplasmic and intranuclear eosinophilic inclusion bodies (arrows). H&E stain, x200 magnification, bar = 50μm.

higher incidence of emaciated animals in this region. Dietary factors may include relative abundance and nutritive quality of prey species [20]. In multiple animals, wood chips, bark, eel grass, kelp and inorganic debris were identified as gastric contents suggesting foraging attempts in sub-optimal habitat. Based on available data and the lack of historic use of a standard nutritional condition protocol, it is difficult to determine whether stranded whales with severe emaciation represent a novel process unique to this UME or if these animals represent

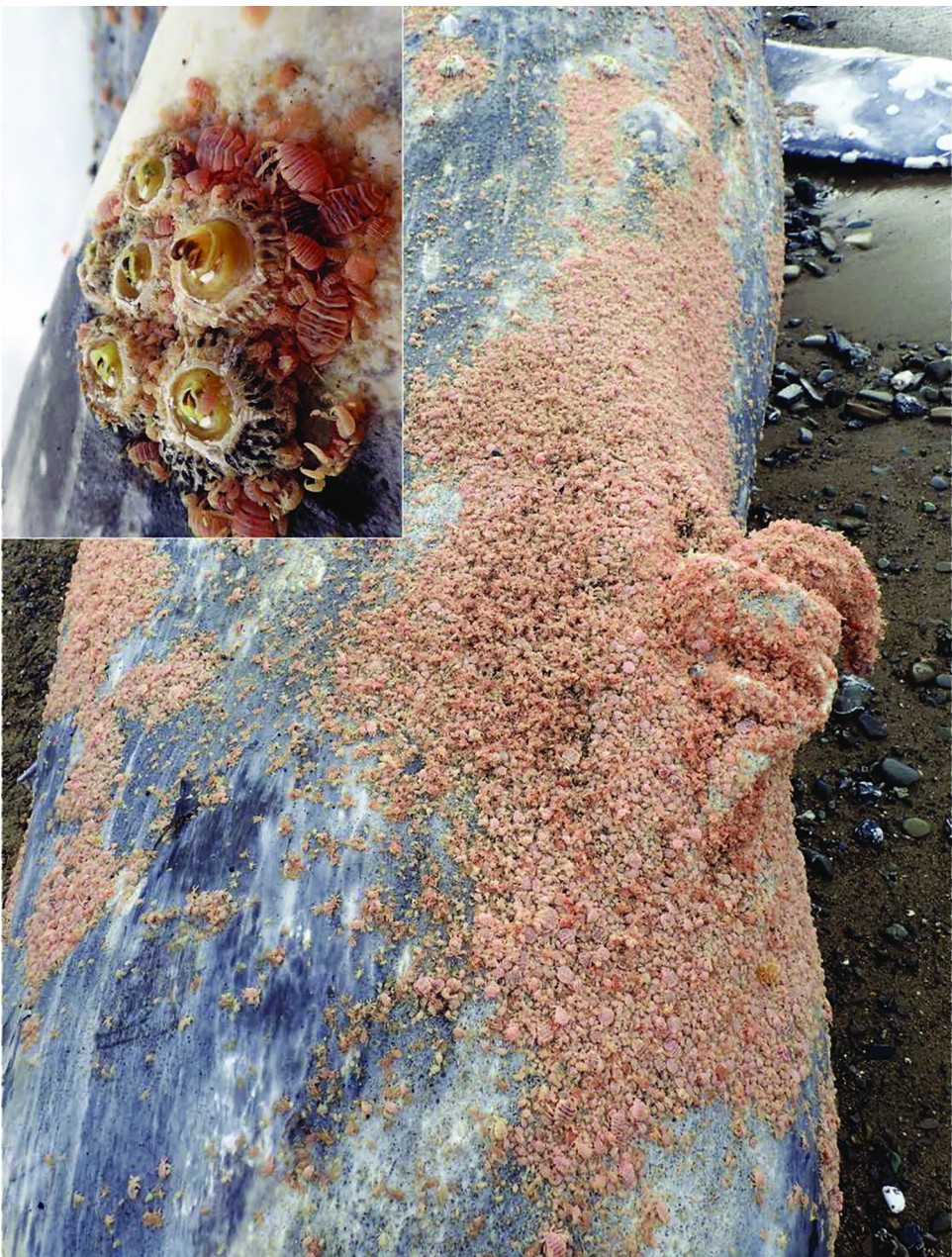

**Fig 7. Emaciated subadult male gray whale with ectoparasitic copepods.** View of genital and perineal area showing massive infestation with cyamids (whale lice). Inset shows closer view of cyamids surrounding attached cirripedes (barnacles) on the skin surface.

typical nutritional status during this phase of the northbound migration in non-UME years as well. More work would be needed to assess historic cases using the nutritional condition protocol developed for this UME for comparison.

In this case series it is difficult to infer a direct association between the salmon coloration of the blubber observed in WA whales and malnutrition and hepatocellular hemosiderosis. The salmon coloration was distinct from jaundice and was most likely due to exogenous pigment deposition, possibly related to prey shifts and consumption of a diet richer in carotenoids. The

subcutaneous edema noted in a few cases of discolored blubber may be secondary to hypoproteinemia, localized trauma, or impaired vascular perfusion as a consequence of live stranding. The accumulation of hepatic iron and atrophy of skeletal muscle in animals stranded in WA and CA substantiated the gross diagnosis of thin or emaciated animals. Hepatocellular hemosiderosis can also be associated with acute infections (due to iron sequestration), chronic inflammation, a maladaptive type syndrome, excessive dietary iron consumption, and other factors. However, significant inflammatory lesions were not observed in these cases.

Vessel strikes were important causes of mortality in this case series. In the 1999–2000 UME, only one confirmed case of vessel strike was documented; however, it is important to note that only three stranded animals were necropsied during the previous UME [31]. Most of the stranded gray whales with gross lesions consistent with vessel strike (Fig 2) were recovered near the shipping channels leading into San Francisco Bay, CA and Puget Sound, WA, which are areas of high relative risk for gray whale vessel strike [28]. Three whales in more remote areas (BC and AK) also had injuries consistent with ship strikes outside of large shipping channels. Some known vessel strike cases reported during the UME were not included in this case study because carcasses were not examined, or due to advanced decomposition at the time they were accessed, as such, the vessel strike numbers reported here are an underestimate. Analysis of all the vessel strike cases during UME and non-UME years could determine if there has been a change in the rate of vessel strike mortality between UME and non-UME years.

We did not observe an association between suboptimal nutritional status and increased propensity for traumatic injury. Rather, extralimital foraging efforts on the northern migration and increased residence time in coastal bays and sounds on the west coast (e.g. San Francisco Bay, CA, Puget Sound, WA and Port of Los Angeles, CA) prolongs the time spent in shipping lanes and may increase the risk of vessel strike, exposure to sound disturbances, and possible disorientation. During this UME, gray whales were observed foraging in San Francisco Bay for periods of time up to a month [55]. A cluster of increased strandings was observed in San Francisco Bay during the 1999/2000 UME and it is possible that migrating whales were stopping to feed in the bay during this event as well [31]. The increased overlap between whales and shipping lanes is concerning from species conservation and individual animal health and welfare perspectives. Ongoing monitoring of gray whale use of San Francisco Bay may be able to predict whether risk is increased during high mortality years.

Rake marks or scars from killer whale attacks are quite common in a variety of baleen whale species. In a photographic study of free ranging blue, humpback, and gray whales in the eastern north Pacific, gray whales had the highest incidence of killer whale rake marks with 42% affected [56]. For fatal predation cases documented in AK, complete loss of the soft tissues (mucocutaneous junction and mucosa) from either or both mandibles and separation of the mandibular symphysis have been reported [47]. In addition, there may also be segmental detachment of the skin and blubber from the underlying fibroelastic sheath with smooth abrupt linear or curvilinear margins and more rarely, abdominal perforations and evisceration [46]. In presumptive non-lethal attacks, there were superficial linear to curvilinear laminated cutaneous furrows, or rake marks, on the appendages, and bite wounds on the margins of the pectoral flippers and flukes. Development of the case definition for killer whale predation as a cause of death for this and future events was an important step towards evaluating these cases consistently throughout the gray whale range. These criteria will help differentiate vessel strike from killer whale predation in AK, where increased vessel traffic through the Bering Strait and pods of transient killer whales that predate gray whale calves occur simultaneously (C. Matkin, pers comm). External examinations of gray whales that did not meet selection criteria for this study, particularly those stranded in OR where full PME was not always feasible, often revealed substantial evidence of killer whale interactions. Future efforts are underway to evaluate all the

gray whale UME cases for signs of killer whale interactions which may illuminate previously undescribed patterns in killer whale behavior.

Carcass decomposition severely limited the ability to conduct comprehensive necropsies on whales with full sampling for histologic examination to rule out primary or secondary involvement of infectious disease, toxins or other disease processes in this UME [32]. Incidental and solitary cases of myocellular sarcocystosis, penile papilloma, pneumonia, enteric cestodiasis and gastric erosions were interesting and valuable observations (Table 8) but did not appear to contribute substantially to the individual health of stranded animals. One of the two calves examined, live stranded in CA and died spontaneously. before the network was able to respond. At necropsy, there were no significant gross lesions, but histopathology revealed mild acute subendocardial and adrenal cortical hemorrhage consistent with agonal catecholamines release (as described in [57]). These lesions may also be associated with septic or hypovolemic shock and other processes but there was no histological evidence for systemic infection. An additional histologic finding was mild bacterial glossitis with cytoplasmic inclusions in the lingual mucosa. Follow up transmission electron microscopy revealed that the cytoplasmic inclusions were composed of keratin filaments and glycogen and were considered incidental findings.

Internal examinations were not pursued on all strandings, however, the gross and microscopic findings we documented did not suggest that infectious agents were a cause of death or stranding during this UME. Based on the lack of inflammation and degree of pulmonary edema, the aspirated debris in lung sections in a few whales was most likely agonal and associated with live stranding. As mortality due to infectious disease can be associated with complex interactions with environmental factors, poor body condition, and etiologic agent, testing of existing samples from this UME for pathogens and biotoxins is on-going, and prospective investigations on rare fresh carcasses should focus on collecting appropriate samples for more advanced analyses.

The northward migration of gray whales along the US West Coast in spring is an annual event that for decades has been documented and reported by generations of biologists and naturalists [18]. Many urban and rural areas of the coast are populated and well-traveled, so that large cetacean strandings are likely observed and reported. In contrast, in AK, carcass discovery is more dependent upon targeted aerial surveys. For this reason, it is unlikely that the large peaks in mortality recorded in 1999–2000 and 2019–2021 are due to increased observational effort. Beach cast and near-shore floating animals represent a small percentage of the overall mortality of large whale populations [58–60]. Factors that impact the stranding event include that the whale dies close enough to shore to eventually strand, that the oceanographic conditions propel the animal to come ashore, that the stranded animal is detected, and that a response team can access the carcass [53]. Because of the extensive nature of the North American coastline, the stark differences in population density, the distribution of response groups, and the often-remote stranding locations of whales, this sample of gray whale strandings is an underestimate of the mortality during this period. Data collection from this study's subset of whales was further limited by decomposition and logistics. The contribution of travel restrictions and the lack of public shore usage associated with the COVID-19 pandemic is unknown; however, in some regions along the coast, responses were curtailed or restricted.

During the UME, the population of eastern north Pacific gray whales declined from approximately 28,000 animals to 14,000 of which 503 were reported stranded along the western seaboard of Mexico, the United States and Canada. A new analysis modeling the long term periodicity of the gray whale calf production, mortality, and nutritional condition with environmental drivers [26] found that periodic reductions in gray whale numbers were associated with prey availability and access to feeding areas in the Arctic. This model fits with malnutrition as the primary finding in investigations of the 1999–2000 and 2019–2021 events. As the Arctic continues to change, the synergy between environmental influences, prey availability, nutritional condition, calf

production, and mortality are increasingly important for understanding the fate of the ENP gray whale population. In particular, the impact of oceanic thermal anomalies in the North Pacific [61], unprecedented loss of winter sea ice and extreme summer sea surface temperatures in the Bering and Chukchi marine shelf ecosystems between 2017 and 2019, and changes in infaunal, benthic and pelagic prey abundance and nutritive quality to the morbidity and mortality of gray whales [19, 20, 25] require further research. The current data and review of the 2019–2021 strandings, as well as comparison with the epidemiologic and necropsy findings from the 1999–2000 UME, support the hypothesis that these events are likely recurring and transitory.

The roles of infectious disease, or intoxication (e.g., biotoxins), as examples of stochastic mortality [62], cannot be ruled out by the current investigation and additional testing of samples will be pursued in the future. The review of gray whale health and disease findings from this investigation, coupled with a recent overview of gray whale pathology literature [32], will better inform future studies of stranded animals.

The data presented here are a small contribution to the overall UME investigation and other teams are working on evaluating environmental and population changes that may be impacting the gray whale population. A better understanding of the manifestations and mechanisms involved in atrophy of blubber during the migration will assist in the determination of normal physiologic atrophy versus malnutrition for this migratory species. Ongoing efforts to document changes in the nutritional status and behavior of free ranging animals, calf recruitment, and foraging behavior will further contribute to understanding the recurrent stranding events of the Eastern North Pacific gray whale population.

In conclusion, the declaration of the UME was invaluable in establishing effective channels of communication and developing and standardizing response protocols across stranding response networks, as well as regional, national, and international governmental partners and agencies which enhanced surveillance and reporting of floating and stranded dead gray whales. The preliminary findings from the subset of 61 whales in this study have shown that malnutrition, vessel strikes, killer whale predation, gear entanglement and entrapment were variably implicated. Our necropsy findings of malnutrition complement the observations of poor body condition in live free ranging whales and low calf recruitment observed during the UME period and support ongoing study of the environmental drivers that are impacting the Eastern North Pacific gray whale population.

## Supporting information

**S1 Appendix. Dead whale nutritional condition protocol_Nov2023.** Gray Whale UME Dead Whale Nutritional Condition Protocol & Table.
(DOCX)

**S2 Appendix. Case definition_ GW KW predation 2023.06.15.** Case Definition: Killer Whale Predation as the Cause of Death in Gray Whales in Alaska.
(DOCX)

## Acknowledgments

All marine mammal stranding activities in the US were conducted under authorization by the NMFS through Stranding Agreements issued to the Marine Mammal Stranding Network Organizations involved in this UME. We acknowledge the assistance of numerous staff and volunteers within the U.S. Marine Mammal Stranding Networks in CA, OR, WA and AK and especially The Marine Mammal Center (TMMC) and the California Academy of Sciences with specific mention of Dr. C. Field, E. Whitmer and J. Isbell (TMMC). We thank C. Matkin for

comments regarding killer predation of gray whales in AK and Dr. S. Moore for invaluable insights. The First Nations communities, Department of Fisheries and Oceans, Fisheries Officers of BC and staff at the BC Animal Health Center, Abbotsford, BC are gratefully acknowledged. Funds for many stranding responses were provided by the Prescott Grants and the Department of Fisheries and Oceans. Tissue collection and processing was conducted under NMFS Stranding Agreements or the Marine Mammal Stranding Response Program Permit number 18768, issued under Section 104 (16 U.S.C. 1374) of the Marine Mammal Protection Act. In MX, the gray whale stranding field work was conducted under the permit from the Secretaría del medio Ambiente y Recursos Naturales No. SGPA/DGVS/013210/18. We thank Steven Swartz, Sergio Martínez A., Lorena Viloria G. and the personnel of Laguna San Ignacio Ecosystem Science Program, PRIMMA-UABCS and the Reserva de la Biósfera El Vizcaino of CONANP. The scientific results and conclusions, as well as any views or opinions expressed herein, are those of the authors and do not necessarily reflect the views of NOAA.

## Author Contributions

**Conceptualization:** Stephen Raverty, Pádraig Duignan, Denise Greig, Jessica L. Huggins, John Calambokidis, Paul Cottrell, Deborah Duffield, Moe Flannery, Frances MD Gulland, Teri Rowles, Deborah Fauquier.

**Data curation:** Denise Greig, Jessica L. Huggins, Kerri Danil, Dalin D'Alessandro, Moe Flannery, Barbie Halaska, Taylor Lehnhart, James Rice, Kate Savage, Kristin Wilkinson, Justin Greenman, Brendan Cottrell, Maggie Martinez.

**Formal analysis:** Stephen Raverty, Pádraig Duignan, Denise Greig, Jessica L. Huggins, Kathy Burek Huntington, Michael Garner, John Calambokidis, Deborah Duffield, Moe Flannery, Frances MD Gulland, Barbie Halaska, Jorge Urbán R., Teri Rowles, Kate Savage, Justin Greenman, Justin Viezbicke, Brendan Cottrell, P. Dawn Goley, Deborah Fauquier.

**Funding acquisition:** John Calambokidis, Paul Cottrell, Teri Rowles, Deborah Fauquier.

**Investigation:** Stephen Raverty, Pádraig Duignan, Denise Greig, Jessica L. Huggins, Kathy Burek Huntington, Michael Garner, John Calambokidis, Paul Cottrell, Kerri Danil, Dalin D'Alessandro, Deborah Duffield, Moe Flannery, Frances MD Gulland, Barbie Halaska, Dyanna M. Lambourn, Taylor Lehnhart, Jorge Urbán R., Teri Rowles, James Rice, Kate Savage, Kristin Wilkinson, Justin Greenman, Justin Viezbicke, Brendan Cottrell, P. Dawn Goley, Deborah Fauquier.

**Methodology:** Denise Greig, Jessica L. Huggins, John Calambokidis, Paul Cottrell, Moe Flannery, Frances MD Gulland, Dyanna M. Lambourn, Teri Rowles, Justin Greenman, Justin Viezbicke, P. Dawn Goley.

**Project administration:** Pádraig Duignan, Denise Greig, Dalin D'Alessandro, Frances MD Gulland, Kate Savage, Justin Greenman, Deborah Fauquier.

**Resources:** John Calambokidis, Kerri Danil, Deborah Duffield, Frances MD Gulland, Barbie Halaska, Dyanna M. Lambourn, Taylor Lehnhart, Teri Rowles, James Rice, Kate Savage, Kristin Wilkinson, Justin Greenman, Justin Viezbicke, Deborah Fauquier.

**Software:** Denise Greig, Barbie Halaska, Brendan Cottrell.

**Supervision:** John Calambokidis, Paul Cottrell, Moe Flannery, Frances MD Gulland, Teri Rowles, Justin Viezbicke, Deborah Fauquier.

**Validation:** John Calambokidis, Brendan Cottrell.

**Visualization:** Brendan Cottrell.

**Writing – original draft:** Stephen Raverty, Pádraig Duignan, Denise Greig, Jessica L. Huggins, Kathy Burek Huntington, Michael Garner, John Calambokidis, Paul Cottrell, Kerri Danil, Dalin D'Alessandro, Deborah Duffield, Moe Flannery, Frances MD Gulland, Barbie Halaska, Dyanna M. Lambourn, Taylor Lehnhart, Jorge Urbán R., Teri Rowles, James Rice, Kate Savage, Kristin Wilkinson, Justin Greenman, Justin Viezbicke, Brendan Cottrell, P. Dawn Goley.

**Writing – review & editing:** Stephen Raverty, Pádraig Duignan, Denise Greig, Jessica L. Huggins, Kathy Burek Huntington, Michael Garner, John Calambokidis, Paul Cottrell, Kerri Danil, Dalin D'Alessandro, Deborah Duffield, Moe Flannery, Frances MD Gulland, Barbie Halaska, Dyanna M. Lambourn, Taylor Lehnhart, Jorge Urbán R., Teri Rowles, James Rice, Kate Savage, Kristin Wilkinson, Justin Greenman, Justin Viezbicke, Brendan Cottrell, P. Dawn Goley, Maggie Martinez, Deborah Fauquier.

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
