## [Decision Letter · Decision Letter 0]

6 Sep 2023

PONE-D-23-22593Gray whale (Eschrichtius robustus) post-mortem findings from 2018-2021 during the Unusual Mortality Event in the Eastern North PacificPLOS ONE

Dear Dr. Raverty,

Thank you for submitting your manuscript to PLOS ONE. After careful consideration, we feel that it has merit but does not fully meet PLOS ONE’s publication criteria as it currently stands. Therefore, we invite you to submit a revised version of the manuscript that addresses the points raised during the review process.

We look forward to receiving your revised manuscript.

Kind regards,

Vitor Hugo Rodrigues Paiva, Ph.D.

Academic Editor

PLOS ONE

Journal Requirements:

2. Thank you for stating the following in the Competing Interests/Financial Disclosure section:

“Dr Kathy Burek owns and operations the Alaska Veterinary Pathology Services, Eagle River, AK and Dr Michael Garner owes the Northwest ZooPath, Monroe, WA”

We note that one or more of the authors are employed by a commercial company: Alaska Veterinary Pathology Services, Eagle River

Reviewers' comments:

Reviewer's Responses to Questions

**Comments to the Author**

1. Is the manuscript technically sound, and do the data support the conclusions?

Reviewer #1: Yes

Reviewer #2: Yes

Reviewer #3: Yes

Reviewer #4: Yes

2. Has the statistical analysis been performed appropriately and rigorously? 

Reviewer #1: N/A

Reviewer #2: Yes

Reviewer #3: No

Reviewer #4: No

3. Have the authors made all data underlying the findings in their manuscript fully available?

Reviewer #1: Yes

Reviewer #2: Yes

Reviewer #3: Yes

Reviewer #4: No

4. Is the manuscript presented in an intelligible fashion and written in standard English?

Reviewer #1: Yes

Reviewer #2: Yes

Reviewer #3: Yes

Reviewer #4: No

5. Review Comments to the Author

Reviewer #1: Review of ‘Gray whale (Eschrichtius robustus) post-mortem findings from 2018-2021 during the Unusual Mortality Event in the Eastern North Pacific’

Thank you for the opportunity to review this manuscript. It is well written and provides detailed , important information on some of the individuals stranded as part of the currently declared UME. The authors should be commended on the detailed step by step appendices provided for defining nutritional state and orca interactions. These will be very valuable for marine mammal science going forwards and will be applicable across species. I have few suggestions but have provided some line-by-line comments below.

I am surprised to see no mention of animal welfare throughout the manuscript, especially in light of some of the causes of death focused on here. Consideration of marine mammal welfare lags behind that of conservation and I believe that careful and considerate mention of it in such a paper as this would be beneficial to marine mammal science and policy. I have included a couple of comments in areas where I believe this could be included with minimal additional interpretation of the findings.

There are some sections in the discussion where additional citations should and could be added, not only to support the points made, but also to direct further reading. For example, the discussion of possible etiologies for hemosiderosis and blubber discoloration. There also seems to be a lack of discussion regarding the entanglement and entrapment cases which I think deserve further attention.

Methods

Line 161-162: Were these human interactions noted based on necropsy of animals or solely on external indicators of potential interaction?

Line 173: Were fetuses considered in the data if they were only found at necropsy of the mother?

Line 193: Due to this, do you think there is bias low numbers of animals with death related to infectious or toxic causes?

Line 197-198: Seeking clarification on this point, it seems that this study focused on these 61 animals only due to them being externally and internally evaluated. This suggests that the other carcasses were not examined in this way, but the next sentence then suggests that there were many others that had been assess extensively enough to suggest human or killer whale interaction. So, how and why were these 61 animals chosen for inclusion in this study?

Line 203: Previous explanation was given on how blubber was used to assess nutritional status, but what was examined in the blubber specifically to assess trauma?

Line 242: What stains were applied to the tissues for histological evaluation?

Results

Line 258: I assume those in other regions were reported by public finding carcasses at sea or beach cast? It might be worth stating this since for AK you have highlighted the main ‘method’ of finding the animals.

Line 262: Here you mention 225 animals stranded in MX but at the start of this paragraph and in Table 1 it states 226?

Line 263: Interested to know if this means that none of the 225 animals stranded in MX underwent any kind of internal gross examination or samples were not sent for histological analysis? Or is this due to the fact that there is a separate study being conducted specifically on the animals in this region? I am surprised as it seems a high proportion of the total strandings to have not been extensively evaluated and included here for overall understanding of findings from this UME.

Line 264 Table 1: As written it is not immediately clear that the number in parentheses refers to the whales with finding reported in this paper, it reads as potentially clarifying that the reported strandings are only those whales with findings being reported. I would suggest re-writing this table legend and others with the same construction to state “Number of whales with findings reported in this paper in parentheses” or something similar:

For example, Table 1. Reported strandings by location and year (17 December 2018 to 31 December 2021). Number of whales with findings reported in this paper in parentheses. MX = Mexico, CA=California, OR=Oregon, WA=Washington, BC=British Columbia, AK=Alaska.

Line 299 Table 4: Misaligned number in 2020 Males for AK.

Line 310: It is surprising that none of the animals initially observed alive underwent PME. Was there a reason for this?

Line 333: Were all these gross findings used for likely cause of death evaluated alongside histology? Did this have any impact on findings when only examined grossly?

Line 341 Figure 2: Images could be of higher resolution (in PDF these are not very clear/some pixelation) but may be an issue with the PDF maker.

Line 394: Does this refer only to external lesions related to vessel strike or orca interactions?

Line 412-423: Not sure why this figure legend appears here for a second time?

Line 426: Could the authors provide further information on this entrapment? And how it contributed to the animal’s death?

Line 455-466: Not sure why this figure legend appears here for a third time?

Line 468: Consider adding ‘likely’ before cause of death, since these results included gross examination only and no histology (thus far, as written).

Discussion

Line 550: Please add ‘as’ after as well and before body condition.

Line 559: This is an example of where consideration of welfare could be adopted.

Line 561: Based on histology of these cutaneous erosions were these ante or post-mortem?

Line 569: This paragraph is an example of where the links among welfare of individuals and conservation of the populations could be highlighted.

Lines 576-579: It would be interesting to know if during this UME there have been any potentially associated out of habitat incidents?

Line 589: Are there any environmental or fisheries data available across this time period to consider modelling such changes alongside the UME?

Line 591: This brings up the question again of whether any animals were considered out of habitat? Or whether this is due to their typical habitat being degraded?

Line 615: Was there any relation with the discoloration of the blubber and unusual gastric contents?

Line 619: It would be interesting to have further discussion of the vessel strike injuries in this paragraph and related welfare and conservation issues.

Line 623: Are these common/typical habitats for these animals to be in?

Line 649: Is there a reference available for this information?

Line 658: Was there any live stranding response effort with this calf and relevant live data collected? This would also be an example for discussion of potential welfare implications associated with this UME.

Line 661: Is it therefore presumed that this was likely related to the live stranding event? If no other underlying causes for stranding appeared present, is it possible that stranding was due to maternal-filial separation?

Line 677: Could you provide some references as suggested further reading

Line 686: Suggestion t include the Moore et al., 2020 paper as a reference for further reading https://www.frontiersin.org/articles/10.3389/fmars.2020.00333/full

Line 689: Can you give any indication as to the likely population impact of this UME?

Line 696: With the current El Niño do you expect UME strandings may begin to increase again?

Reviewer #2: Review comments

This manus describes necropsy findings and death causes of 61 stranded gray whales Eschrichtius robustus along the west coast of North America and review the number of strandings of 503 gray whales geographically, seasonally, and due to life stage. Mortality causes are related to human interactions and carrying capacity is mentioned as a cause of increased mortality in gray whales. The paper is descriptive and contains no tested hypotheses but contributes valuable knowledge of mortalities in gray whales although based on relatively few individuals.

In general, the manus is well written. However, considering the descriptive and relatively few necropsy data and that only theoretical thoughts of possible environmental cause for the decline in population and calf production of gray whales the manus should be shortened. Some may be done by omitting sentences with spoken language, by avoiding repetitions especially in the discussion and deleting adjectives which do not contribute with exact knowledge.

Some suggestions and questions:

Line 54-83 Abstract is too long and should be no longer than 300 words, reduce some of the background information and focus on purpose and results the review.

95: “greatest” – I would prefer “highest.”

99: “genetic studies have estimated higher numbers existed through evolutionary time” how high numbers would be interesting to compare with later models?

104-106: “Transformative ecosystem changes have occurred in the summer Bering Strait feeding

grounds [14] with reported local declines in prey abundance and quality [15] which may have a

direct impact on gray whale bioenergetics, fecundity, health, and homeostasis [16]”. Could be written shorter and contain the same information “Changes in the summer Bering Strait feeding

grounds with reported local declines in prey abundance and quality may influence gray whales’ bioenergetics, fecundity, health, and homeostasis [14,15,16]”.

105: What are gray whales basic diet?

107: How is calf production measured- by random or systematic sightings?

264: “Reported strandings (including whales with findings reported in this paper)”

284: This ads up to 544 is it because calves and fetuses are not included?

320: Gross findings- I would prefer -External findings

531-535: Much is repetition- may even be deleted.

717-721: “UME was invaluable in establishing effective channels of communication and developing and standardizing response protocols across stranding response networks, as well as regional, national, and international governmental partners and agencies which enhanced surveillance and reporting of floating and stranded dead gray whales”. Delete this section. To me this is not a biological conclusion and should not be part of the conclusion. After the biological conclusion it could be mentioned that in future protocols should be standardised and national and international neworks improved.

Reviewer #3: Thank you for letting me reviews this paper. This manuscript studies a very important topic, which is the relevant information that the identification and monitoring of Unusual Mortality Event (UME) are important as they allow scientists and conservation organizations to collect data on possible causes of mortality, assess the impact on the gray whale population, and take measures to protect this endangered species.

The study focuses on the analysis of post-mortem results of gray whales, which means it examines the characteristics and observations made after the death of these animals. The mentioned unusual mortality event refers to a period when an abnormally high number of gray whales died in the Northeast Pacific region.

The purpose of this study is therefore to analyze these post-mortem results in order to better understand this phenomenon of unusual mortality in gray whales. This could help determine the possible causes of these deaths and develop conservation and protection measures for this endangered species.

I have directly written my comments and some corrections in the manuscript.

Reviewer #4: The authors present an analysis of the recent and ongoing gray whale unusual mortality event (UME) in the Eastern North Pacific. They compare this UME to a previous UME and highlight a south to north gradient in potential causes of mortality. I think this work is interesting, and the potential approximately 23 periodicity in these events is especially interesting and insightful to future conservation efforts for this species.

However, the manuscript as it currently stands is poorly written. There are many verbose sections that are written in the passive voice that need to be converted to the active voice. UME is referenced in the text before it is defined. Portions of the text feel like they conflict each other, especially when discussing what samples are considered regarding orca-related mortality. The text needs to be consolidated and written concisely.

Furthermore, the results are quickly lost in the jargon of the manuscript, which currently feels like it was written as an internal report rather than a broader audience. It is very difficult to follow along with the conclusions drawn by the authors as a result. Most of the results describe diseases and conditions that are not understood by a general audience and there is no attempt to define most of, if not all, of these terms. These intricate details may be better suited for a supplement.

Most of the results are simply regurgitating the tables presented (some of which are duplicated in my copy of the manuscript). I feel like 1) the results text should not simply restate the tables and 2) that the results would be easier to comprehend using maps to illustrate the spatial distribution of causes of mortality. This would better help illustrate the S-N gradient the authors discuss. The current figures (images of biopsy samples) are not useful to a broader audience without a background in pathology or marine mammals, and also don’t translate well to grayscale.

There is also no attempt to perform any statistical analysis on these data. The authors state that they do not have enough sample points to perform statistical analysis but do not perform any power calculations to back this up. Statistical analysis should be performed and would strengthen the manuscript considerably.

I think the most interesting part of this paper is the potential 23 periodicity in UME and unfortunately, this or any other environmental drivers of the UME are not discussed. I understand the division of labor across a diverse research team such as this one and the desire to publish results sooner rather than later. However, I think the manuscript would be strengthened significantly if these mortality results were correlated to the environmental variables in the same manuscript.

Based on these comments, I am recommending to reject this manuscript and encourage the authors to resubmit their manuscript when the environmental analysis is complete and can be added to these results.

6. PLOS authors have the option to publish the peer review history of their article (what does this mean?). If published, this will include your full peer review and any attached files.

Reviewer #1: No

Reviewer #2: No

Reviewer #3: No

Reviewer #4: No

---

## [Author Response · Author response to Decision Letter 0]

8 Nov 2023

Reviewer #1: Review of ‘Gray whale (Eschrichtius robustus) post-mortem findings from 2018-2021 during the Unusual Mortality Event in the Eastern North Pacific’

Thank you for the opportunity to review this manuscript. It is well written and provides detailed , important information on some of the individuals stranded as part of the currently declared UME. The authors should be commended on the detailed step by step appendices provided for defining nutritional state and orca interactions. These will be very valuable for marine mammal science going forwards and will be applicable across species. I have few suggestions but have provided some line-by-line comments below.

I am surprised to see no mention of animal welfare throughout the manuscript, especially in light of some of the causes of death focused on here. Consideration of marine mammal welfare lags behind that of conservation and I believe that careful and considerate mention of it in such a paper as this would be beneficial to marine mammal science and policy. I have included a couple of comments in areas where I believe this could be included with minimal additional interpretation of the findings.

There are some sections in the discussion where additional citations should and could be added, not only to support the points made, but also to direct further reading. For example, the discussion of possible etiologies for hemosiderosis and blubber discoloration. There also seems to be a lack of discussion regarding the entanglement and entrapment cases which I think deserve further attention.

Thank you for your comments, they are informative and have been addressed in the responses below and added to the text of the manuscript. The lines identified in the responses refer to the revised manuscript in Track Changes. 

Methods

Line 161-162: Were these human interactions noted based on necropsy of animals or solely on external indicators of potential interaction?

Human interactions were based on internal and external examinations. Additional details provided in lines 164-165. 

Line 173: Were fetuses considered in the data if they were only found at necropsy of the mother?

Other than a single fetus that was recovered from during the Unusual Mortality Event (UME) that was described in this manuscript and a separate publication, (Cottrell et al, 2022), there were no pregnant adult females necropsied. This likely represents the stage of fetal development at the time of southern migration of the whales. 

Reference 41: Cottrell, P, Cottrell, B, Dowdall, T, Hoyland, Al, Hoyland, AS, and Raverty, S. Recovery of a mid-gestational gray whale (Eschrichtius robustus) fetus near Tofino, British Columbia. Aquatic Mammals. Aquatic Mammals 2022, 48(6), 626-633, DOI 10.1578/AM.48.6.2022.626

Line 193: Due to this, do you think there is bias low numbers of animals with death related to infectious or toxic causes?

Due to the remoteness of many strandings and delays in arranging travel and logistics to conduct a necropsy, many animals presented with moderate to advanced autolysis (code 3.5-4), which precluded ancillary diagnostic studies to screen for infectious disease. Rather than stage of fetal development or age of the animal, post mortem decomposition may have been among the most significant limiting factors in screening for infectious disease. Harmful algal bloom toxin screening was performed in a few cases, but feces, urine or stomach contents were not consistently sampled or screened; additionally, it is not generally known what toxin levels mean in fluids from large whales. The designation of biotoxin as a cause of death is based on timing and location of stranding, toxin levels in prey species and the environment with documented toxin blooms in the same time and place of the strandings. Because we do not think toxins were the cause of this mortality event, we excluded the partial results from our subset of examined whales; once the UME is closed, a separate summary manuscript including HABs levels is planned. Ultimately, we don’t know if we are biased against infectious or toxic causes or if gray whales are resilient to the diseases processes, when compared to other cetaceans. Responding to fresh gray whale carcasses is a priority throughout their range. Unfortunately, this is a rare occurrence. 

Line 197-198: Seeking clarification on this point, it seems that this study focused on these 61 animals only due to them being externally and internally evaluated. This suggests that the other carcasses were not examined in this way, but the next sentence then suggests that there were many others that had been assess extensively enough to suggest human or killer whale interaction. So, how and why were these 61 animals chosen for inclusion in this study?

With a decline from 26,000-27,000 animals to approximately 14,000 during the span of the UME, level A data was compiled for 501 stranding events. In these cases, the lay public, biologists or researchers would have recorded the geographic location, signalment (age, sex), date, and in some cases, photographs were obtained. However, in many cases there were insufficient data to assign a specific cause of death. Without an internal and internal examination, we could not confirm whether some observed or photographed injuries may have been incurred ante versus post mortem (impact with the substrate) or differentiate attempted predation or post mortem scavenging. Post mortem decomposition, particularly with widespread skin sloughing and seepage of blubber oil impeded and in cases with more advanced autolysis, precluded gross assessment of internal and external lesions. The case definition we developed (and agreed upon as a group) for inclusion as a case in this study included gross internal and external examination of the carcass and/or microscopic evaluation of the tissues for evidence of hemorrhage (vital response), blunt force injury and other disease entities. 

Lines 197-208 in the revised version with Track Changes explain our inclusion criteria.

Line 203 was reworded to “The cases discussed in this paper are for those carcasses where an external and internal gross necropsy was performed and tissues for histopathology were evaluated.”

The following sentence in line 239 to 245 was reworded to “Criteria for inclusion in this study were a full gross PME and, for most cases, sampling for histopathology. Exclusions were based on incomplete PMEs or microscopic assessment of sampled tissues, even if the case was confirmed by external examination as a probable or suspect case of human or killer whale interaction. As such, the numbers reported herein are a minimum and represent a subset of the full UME response and investigation”.

Line 203: Previous explanation was given on how blubber was used to assess nutritional status, but what was examined in the blubber specifically to assess trauma?

Trauma was assessed by evidence of hemorrhage, laceration or incision in soft tissues and fractures of skeletal elements. On gross examination, this was inferred by red discoloration of the blubber with possible edema (tissue fluid accumulation) and often featured subcutaneous or hypodermal hemorrhage and possible hematoma formation, particularly below the fibroelastic sheath. In some cases of cryptic trauma, there may be no discernible external lesions with hemorrhage limited to the blubber and subcutaneous tissue. Microscopically, trauma was assessed by a combination of tissue necrosis, hemorrhage, edema and nature and extent of the inflammatory infiltrate. With acute hemorrhage, red blood cells would predominate and with resolution (in nonlethal cases) hemosiderin laden macrophages admixed with varying amount of other more chronic inflammatory infiltrates predominate. 

In lines 250-255, an additional sentence describing the gross features of trauma has been added for clarification. Trauma was inferred by evidence of epidermal lacerations hemorrhage, edema and necrosis or maceration of skeletal muscle, and skeletal fractures. 

Line 242: What stains were applied to the tissues for histological evaluation?

The tissues were routinely stained with Hematoxylin and Eosin (H&E). An additional sentence has been added in line 298, “ Tissues were stained with hematoxylin and eosin and when indicated, Tort’s Gram stain, Periodic Acid Schiff (PAS) and Perl’s stain (for iron) and reviewed by a veterinary pathologist.”

Results

Line 258: I assume those in other regions were reported by public finding carcasses at sea or beach cast? It might be worth stating this since for AK you have highlighted the main ‘method’ of finding the animals.

In lines 324-325, a statement has been added to clarify the difference between reporting or detecting dead whales in different geographic regions. The sentence now reads “In contrast to BC, WA, OR, CA and MX where dead whales were reported by the public, biologists, fisheries officers, First Nations or Indigenous community members, most of the reports for AK were from aerial surveys or opportunistic aerial sightings of floating or beach cast animals.”

Line 262: Here you mention 225 animals stranded in MX but at the start of this paragraph and in Table 1 it states 226?

The number in the table is correct and the number in the text in line 330 has been revised from 225 to 226. Thank you

Line 263: Interested to know if this means that none of the 225 animals stranded in MX underwent any kind of internal gross examination or samples were not sent for histological analysis? Or is this due to the fact that there is a separate study being conducted specifically on the animals in this region? I am surprised as it seems a high proportion of the total strandings to have not been extensively evaluated and included here for overall understanding of findings from this UME.

Yes, this is a deficiency. In general, the carcasses are in remote areas where it is difficult to mobilize gear and people to conduct a full exam. There are dedicated surveys to count and measure them, but these surveys are not frequent enough to find fresh carcasses. We would all like to collect more information from the wintering grounds (particularly the near term fetal and neonate mortality cohorts). Necropsy gear has been sent from the US to Mexico. There were three necropsies conducted, but as their results did not change our conclusions, our Mexican co-authors opted not to include them in this paper.

Line 264 Table 1: As written it is not immediately clear that the number in parentheses refers to the whales with finding reported in this paper, it reads as potentially clarifying that the reported strandings are only those whales with findings being reported. I would suggest re-writing this table legend and others with the same construction to state “Number of whales with findings reported in this paper in parentheses” or something similar:

For example, Table 1. Reported strandings by location and year (17 December 2018 to 31 December 2021). Number of whales with findings reported in this paper in parentheses. MX = Mexico, CA=California, OR=Oregon, WA=Washington, BC=British Columbia, AK=Alaska.

Line 333-233, thank you. We added “The number in the parentheses is the subset of whales with findings reported in this paper.” to the captions of Tables 1-5.

Line 310: It is surprising that none of the animals initially observed alive underwent PME. Was there a reason for this?

There are multiple limitations, but the most important was coordinating and getting the whale and people to a location where we were permitted to perform a necropsy. Roadblocks are listed lines 229-235, but often we are desperately trying to get to the carcass or figure out where it will float to next. As described in Michael Moore’s paper, in some instances a fresh dead whale has been reported, but we think they sink initially, then we have to wait for the carcass to bloat, before it floated and could be relocated. 

Line 333: Were all these gross findings used for likely cause of death evaluated alongside histology? Did this have any impact on findings when only examined grossly?

Yes, we included both gross and microscopic evaluation of the tissues included in the assignment of cause of death. In some cases, where catastrophic injuries were apparent, a cause of death could be determined without histopathology.

A sentence has been added to line 309. “A group of veterinary pathologists and field personal experiences with post mortem examination of large cetaceans reviewed each of the 61 cases to assign a cause of death. Where discrepancies occurred, the field personnel involved with the necropsy and tissue sampling were consulted and photographs at the time of necropsy reviewed.” 

Line 341 Figure 2: Images could be of higher resolution (in PDF these are not very clear/some pixelation) but may be an issue with the PDF maker.

The images and figures have been reformatted to TIF files and the quality should be improved. Thank you!

Line 394: Does this refer only to external lesions related to vessel strike or orca interactions?

The statement refers specifically to the cause of death by starvation. Suboptimal nutritional condition may be associated with a variety of disease processes including poor quality or low caloric dietary intake, intercurrent or pre-existing disease, particularly within the gastrointestinal (lack of nutritional resorption) or urogenital (excess protein excretion) and other factors (physiologic, associated with the prolonged migration, reproductive status and others). In this case series, there were no apparent lesions which may have contributed to the emaciation. 

The sentence in line 503-505 has been reworded to 

“The primary cause of death was starvation in 16 of 18 (89%) of the emaciated whales, with no apparent pre-existing pathology that would have contributed to suboptimal nutritional condition.”. 

Line 412-423: Not sure why this figure legend appears here for a second time?

The repeated text has been removed from lines 444 and 455. Thank you

Line 426: Could the authors provide further information on this entrapment? And how it contributed to the animal’s death?

The animals typically swim under a pier or wharf between pilings and are unable to escape. Animals have been observed to become agitated with repeat efforts to escape. A specific cause of death has not yet been determined for these mortalities, but we hypothesize that an exertional or capture type myopathy (related to metabolic acidosis and low oxygen levels, hypoxia) may be contributing factors. 

A statement has been added to line 541-543 “The proximate cause of death of this animal has not yet been fully resolved but may be associated with exertional or capture type myopathy.” And references 48 and 49 added. 

Line 455-466: Not sure why this figure legend appears here for a third time?

The legend text has been removed from line 494-505, thank you 

Line 468: Consider adding ‘likely’ before cause of death, since these results included gross examination only and no histology (thus far, as written).

Histopathology would have been conducted in these cases, but with post mortem decomposition or incomplete tissue sampling, determination of a specific cause of death may have been hindered. “Likely” has been added. Thank you

Discussion

Line 550: Please add ‘as’ after as well and before body condition.

added, thank you

Line 559: This is an example of where consideration of welfare could be adopted.

Thank you, a statement has been added to line 725-726 

“In live strandings, animal welfare considerations and consultation with clinical veterinarians is warranted. “ A citation has been added to the text, reference 53, Boys et al, 2022.

Line 561: Based on histology of these cutaneous erosions were these ante or post-mortem?

Based on the historical information, clinical signs and gross findings, these ulcers were considered antemortem. Not all ulcers were sampled for histologic review and in many cases, defects in the skin result in osmotic breach, imbibed water, and bacterial overgrowth which may hinder microscopic evaluation of the tissues. The histopathology typically features lysis of the exposed dermis, which can be difficult to differentiate between ante and post mortem. 

Line 569: This paragraph is an example of where the links among welfare of individuals and conservation of the populations could be highlighted.

The following statement has been added to line In live stranding situations, animal welfare considerations and consultation with appropriately experienced clinical veterinarians is warranted . Boys et al reference has been added to line 723, reference 53.

Lines 576-579: It would be interesting to know if during this UME there have been any potentially associated out of habitat incidents?

There have not been any out of habitat incidents associated with this UME. It is normal for these animals to travel close to shore, but spending days to weeks in SF Bay to forage is new behavior which puts them in the shipping channels for extended periods of time. 

Interestingly there was an increase in gray whale in SF Bay in 1999/2000 as well as this UME. We have added information about this lines 790-801 with references 30 and 54 added. 

Line 589: Are there any environmental or fisheries data available across this time period to consider modelling such changes alongside the UME?

Yes, and looking at the longer gray whale stranding record. A recent manuscript has just been published (Stewart et al 2023 Boom-bust cycles in gray whales associated with dynamic and changing Arctic conditions). A team of investigators modelled the effects of sea ice loss, duration of access to foraging areas, and benthic fauna abundance and biomass and demonstrated a significant association with whale mortality and reproductive output. 

We purposely focused on the pathology in this paper, and work to combine nutritional condition and cause of death information in UME and non-UME years will be much bigger effort.

Line 591: This brings up the question again of whether any animals were considered out of habitat? Or whether this is due to their typical habitat being degraded?

The distribution of reported dead whales is along the migratory corridor. There have been anecdotal reports of individual adult or cow calf pairs foraging in shallow mud flats, but not associated with mortality. In the past, individual gray whales have migrated up the Sacramento River in California but there have been no out of habitat cases documented during this UME (please refer to response above and lines 790-801 in the track changes manuscript. 

Line 615: Was there any relation with the discoloration of the blubber and unusual gastric contents?

There were no consistent relations between gastric contents and discoloration of the blubber. With discolorations of this type, we consider either endogenous or exogenous pigments. Endogenous pigments may include jaundice or another metabolic byproduct. In this case, we think exogenous pigments are more likely, possibly related to prey shifts and increased consumption of crustaceans (carotenoids); however, due to inconsistencies in post mortem state, examination of gastric contents, stranding locations, and body condition, it was difficult to infer a specific mechanism for the discoloration. 

Line 619: It would be interesting to have further discussion of the vessel strike injuries in this paragraph and related welfare and conservation issues.

This is an important consideration and, we have added text to highlight the need for animal welfare considerations with the live animal strandings and the increased risk of vessel strike if malnourished animals are spending more time foraging within high traffic areas along the migration route.

We are very concerned about the animal welfare issues, this year (2023 - beyond the timeline of this paper), a vessel strike whale was observed alive and injured in San Francisco Bay prior to stranding dead. That is helping us to mobilize various interests within the bay to think about how to avoid hitting whales. 

Line 623: Are these common/typical habitats for these animals to be in?

Yes, the gray whales transit a series of shipping lanes in more populated areas along the coast San Francisco, Los Angeles, and Seattle. They hug the coastline (maybe to avoid killer whales?) and shipping lanes going in and out of ports are a concern.

Line 649: Is there a reference available for this information?

This statement was added at the request of a coauthur (JR) in Oregon. Although only a small number of dead whales are reported in OR, there were multiple individuals with gross evidence of rake marks, attributed to killer whales. Similar observations have more recently been observed in the ocean trench off Santa Barbara, CA. Unfortunately, these reports are anecdotal and to the best of our knowledge, have not yet been published. As part of cohort of animals that presented with evidence of killer whale predation, there has been some discussion of drafting manuscript, but there is still insufficient necropsy data to confirm ante versus post mortem attacks and whether evasive flight from killer whales may result in live strandings and subsequent deaths. 

Line 658: Was there any live stranding response effort with this calf and relevant live data collected? This would also be an example for discussion of potential welfare implications associated with this UME.

Yes there was, however, the calf was dead by the time responders arrived.

Line 661: Is it therefore presumed that this was likely related to the live stranding event? If no other underlying causes for stranding appeared present, is it possible that stranding was due to maternal-filial separation?

Yes, this may be a consideration, but the separation may be related to maternal loss, neglect or mismothering. Without direct field observation it is difficult to specify a cause of the stranding. Lack of colostral consumption and nursing may have resulted in hypoglycemia and with generalized weakness or debilitation, the neonate or calf may have live stranded and subsequently died. 

Line 677: Could you provide some references as suggested further reading

The references below provide a good historical perspective. Both references have been added to the text and list. 

Eguchi, T, Lang, AR, Weller, DW. Abundance and migratory phenology of eastern North Pacific gray whales 2021/2022. U.S. Department of Commerce, NOAA Technical Memorandum NMFS-SWFSC-668. 2022. https://doi.org/10.25923/x88y-8p07

Also the recent paper from Steward et al as been added:

Stewart JD, Joyce TW, Durban JW, Calambokidis J, Fauquier D, Fearnbach H, et al. Boom-bust cycles in gray whales associated with dynamic and changing Arctic conditions. Sci. 2023; 382, 207-211

Line 686: Suggestion t include the Moore et al., 2020 paper as a reference for further reading https://www.frontiersin.org/articles/10.3389/fmars.2020.00333/full

Reference added, thank you 

Line 689: Can you give any indication as to the likely population impact of this UME?

Please refer to comments provided for Lines 871-882. Gray whales have recovered from similar mortality events before, but there is a lot of uncertainty going forward because of the rapidity of change in the Arctic.

Line 696: With the current El Niño do you expect UME strandings may begin to increase again?

Please refer to comments provided for Lines 871-882. Honestly, we’re in new territory with all the changes in the Arctic, plus marine heatwaves, plus a shift in the Pacific Decadal Oscillation from cold regime to warm regime. We don’t know whether the current El Niño will exacerbate or mitigate any of these impacts on gray whale environment and prey.

 

Reviewer #2: Review comments

This manus describes necropsy findings and death causes of 61 stranded gray whales Eschrichtius robustus along the west coast of North America and review the number of strandings of 503 gray whales geographically, seasonally, and due to life stage. Mortality causes are related to human interactions and carrying capacity is mentioned as a cause of increased mortality in gray whales. The paper is descriptive and contains no tested hypotheses but contributes valuable knowledge of mortalities in gray whales although based on relatively few individuals.

In general, the manus is well written. However, considering the descriptive and relatively few necropsy data and that only theoretical thoughts of possible environmental cause for the decline in population and calf production of gray whales the manus should be shortened. Some may be done by omitting sentences with spoken language, by avoiding repetitions especially in the discussion and deleting adjectives which do not contribute with exact knowledge.

Some suggestions and questions:

Line 54-83 Abstract is too long and should be no longer than 300 words, reduce some of the background information and focus on purpose and results the review.

The text in lines 54-83 provide a historic perspective of the epidemiology and pathologic findings of a past UME, we have included this information to provide some context about the current study. If possible, we would like to retain this information, but if requested by the editor, can remove the content. As far as we can tell, PLOSONE does not limit abstracts to 300 words. Thank you for your comment and recommendation. 

95: “greatest” – I would prefer “highest.”

In line we have changed greatest to largest. Thank you, now line 103 in Track Changes 

99: “genetic studies have estimated higher numbers existed through evolutionary time” how high numbers would be interesting to compare with later models?

A separate modeling manuscript on gray whale abundance which takes long term stranding data into consideration is scheduled to publish this month. In this review, we are focused specifically on describing the signalment (age, sex, etc), demographics and post mortem findings (cause of death) of stranded animals. 

104-106: “Transformative ecosystem changes have occurred in the summer Bering Strait feeding

grounds [14] with reported local declines in prey abundance and quality [15] which may have a

direct impact on gray whale bioenergetics, fecundity, health, and homeostasis [16]”. Could be written shorter and contain the same information “Changes in the summer Bering Strait feeding

grounds with reported local declines in prey abundance and quality may influence gray whales’ bioenergetics, fecundity, health, and homeostasis [14,15,16]”.

The sentence has been replaced with the suggested text in lines 114-116. Transformative has been retained at the start of the sentence due to the profound changes which have occurred in the region.

105: What are gray whales basic diet?

The primary prey are benthic and epibenthic invertebrates, including amphipods, mysids, crustaceans and plankton. The diet will vary by geographic location and in the north Pacific amphipods and isopods are the primary prey 

In lines 119-121, the following statement has been added for additional background information and citations 18, 23, and 24 have been added. Thank you.

The diet varies by geographic location and whether gray whales forage in shallow mud flats or deeper waters and consists primarily of benthic and epibenthic invertebrates, including amphipods, crustaceans, mysids, and plankton.

107: How is calf production measured- by random or systematic sightings?

Systematic sightings, please refer to citation 18 and 60, which has been added to the text, line 130. Thank you. 

264: “Reported strandings (including whales with findings reported in this paper)”

Yes. The table refers specifically to data derived by this study. 

284: This ads up to 544 is it because calves and fetuses are not included?

The total in the table is 503 and includes 2 whales that stranded in December 2018. There may be some confusion related to the numbers in the parentheses, which reflect the subset of animals that were necropsied (post mortem exam). Per the request of reviewer 1, text has been added to line 353-354 in Track Changes for clarification. Thank you 

320: Gross findings- I would prefer -External findings

We respectfully disagree, external findings would be limited to only the appearance of the skin and confirmation of the carcass with some indication of nutritional condition. Gross findings infers both internal and external observations, which are critical to assess ante versus postmortem changes (vital response such as hemorrhage and edema to injuries which may manifest as internal lesions and not readily apparent on external examination). We retained the term Gross findings because it is standard anatomic pathology terminology, and this paper focuses on descriptive pathology. 

531-535: Much is repetition- may even be deleted.

We appreciate the comment provided by the reviewer, but respectfully disagree with removal of this statement. The paragraph is intended to provide a summary of the study findings, which are then addressed in more detailed throughout the remainder of the Discussion. 

717-721: “UME was invaluable in establishing effective channels of communication and developing and standardizing response protocols across stranding response networks, as well as regional, national, and international governmental partners and agencies which enhanced surveillance and reporting of floating and stranded dead gray whales”. Delete this section. To me this is not a biological conclusion and should not be part of the conclusion. After the biological conclusion it could be mentioned that in future protocols should be standardised and national and international neworks improved.

The first sentence from line 783 has been moved to the end of the paragraph 

 

Reviewer #3: Thank you for letting me reviews this paper. This manuscript studies a very important topic, which is the relevant information that the identification and monitoring of Unusual Mortality Event (UME) are important as they allow scientists and conservation organizations to collect data on possible causes of mortality, assess the impact on the gray whale population, and take measures to protect this endangered species.

The study focuses on the analysis of post-mortem results of gray whales, which means it examines the characteristics and observations made after the death of these animals. The mentioned unusual mortality event refers to a period when an abnormally high number of gray whales died in the Northeast Pacific region.

The purpose of this study is therefore to analyze these post-mortem results in order to better understand this phenomenon of unusual mortality in gray whales. This could help determine the possible causes of these deaths and develop conservation and protection measures for this endangered species.

I have directly written my comments and some corrections in the manuscript.

Thank you for your comments and editorial suggestions on the manuscript. We have tried our best to refer to the lines identified in the original version, but the responses below indicate the line number in the revised version.

Abstract, lines 74-83. We respectfully disagree with removal of this text. Although it is not a direct reflection of findings from our paper, the text provides important historic context in terms of population trends, a prior unusual mortality event and potential contributing factors to the decline in the number of eastern Pacific gray whales. 

Introduction

Line 91, the added before western, thank you

Line 95, have has been changed to has, thank you 

Line 102, comma inserted after recovery

Line 105, thought to be, has been replaced was probably less than 30,000… 

Line 106, that placed before higher. 

Line 109 has been revised. The 2015/2016 eastern Pacific gray whale abundance estimate was 26,960 individuals, which declined to 20,580 in 2019/2020 and further to 16,650 in 2021/2022 [13]. 

Lines 114-116, text revised, thank you, but some elements from the original sentence have been retained. We hope this may be acceptable. 

Transformative changes in the summer Bering Strait feeding grounds [14] with reported local declines in prey abundance and quality [15], which may have a direct impact on gray whale bioenergetics, fecundity, health, and homeostasis [16].

Lines 114 and 115 comma added after [14] and [15]

Line 122, 23 year inserted for 23- year

Line 127, content of parentheses revised to i.e., fewer ice-free days

Line 130 comma inserted after [20,13], please note reference list expanded and the citation numbers have been updated

Line 133, the impacts replaced to impacts, thank you 

Line 136, for clarification, the start of the sentence has been rephrased to These traumatic injuries may be .. we would to retain this statement in the text. 

Line 139, north replaced with North thank you 

Line 150, comma inserted after North America, please note that per the request of reviewer 4, this text has been removed. 

Line 151, a replaced with the, please see comment for line 150

Line 152, due to the small number of examined animals and anecdotal reporting, we cannot assign a percentage for this finding. 

Line 155, the sentence describes the pathologic findings in 3 animals, one with encephalitis, one with heavy intestinal parasitism and another with high domoic acid levels. If possible, we would like to retain this sentence as it provides a brief overview of prior diagnostic findings. 

Line 153, an has been removed. 

Line 153, 155, reference changed to 25 and now revised to 30. 

Line 157, the list of environmental and disease processes provided in this sentence (from citation listed as 25), are general considerations, but not conclusive findings from the 3 examined dead animals during the UME. This statement was included in the text as baseline considerations for disease investigations with stranded animal. We would like to retain the sentence as stated in the text as it reflects the cited publication. 

Line 294, or replaced with comma 

Thank you for the comments about missing statistics. We have re-worked the Results section extensively to avoid redundancy between text and tables and deleting statements that would have required statistical testing. We should not have made those statements as this is a subset of data. Our intent is to describe out dataset and put it in context, but not to make any assertions about differences between years, regions, age classes, or other factors. Table 6, Title changed to Significant causes of death, the summary removed from Causes of Death and totals changed to total with an additional row added under undetermined, the X’s were placeholders and have been revised. Table 7 refers to the nutritional condition of stranded animals. Thank you for your comments and suggestions. 

Lines 388-390 Table 6 revisions and Line 465, Table 6 removed as it was a duplicate 

Line 916 here have found has been removed. Thank you 

Reviewer #4: The authors present an analysis of the recent and ongoing gray whale unusual mortality event (UME) in the Eastern North Pacific. They compare this UME to a previous UME and highlight a south to north gradient in potential causes of mortality. I think this work is interesting, and the potential approximately 23 periodicity in these events is especially interesting and insightful to future conservation efforts for this species.

However, the manuscript as it currently stands is poorly written. There are many verbose sections that are written in the passive voice that need to be converted to the active voice. UME is referenced in the text before it is defined. Portions of the text feel like they conflict each other, especially when discussing what samples are considered regarding orca-related mortality. The text needs to be consolidated and written concisely.

With the constructive comments provided by reviewers 1, 2 and 3, portions of the text in the manuscript have been refined. We have also revised portions of the text from a passive to active voice. We have deleted extraneous information and combined some previously redundant sections: we went from 10, 245 words to 7800 words.

Unusual mortality event (UME) has been corrected in lines 113 and 152. 

For additional corrections, please refer to the bottom of this text.

Furthermore, the results are quickly lost in the jargon of the manuscript, which currently feels like it was written as an internal report rather than a broader audience. It is very difficult to follow along with the conclusions drawn by the authors as a result. Most of the results describe diseases and conditions that are not understood by a general audience and there is no attempt to define most of, if not all, of these terms. These intricate details may be better suited for a supplement.

We respectfully disagree with your assessment of the use of scientific and pathologic terms in the manuscript. For those scientists, veterinarians and pathologists interested in marine mammal health, familiarity with these terms will help understand disease processes in gray whales and other marine mammal species. As many disease processes described in this paper have not previously been well documented in gray whales, extrapolation of nomenclature from terrestrial and other marine mammal species informs how to investigate disease and provides some insights into differentials and diagnostic strategies. 

Most of the results are simply regurgitating the tables presented (some of which are duplicated in my copy of the manuscript). I feel like 1) the results text should not simply restate the tables and 2) that the results would be easier to comprehend using maps to illustrate the spatial distribution of causes of mortality. This would better help illustrate the S-N gradient the authors discuss. The current figures (images of biopsy samples) are not useful to a broader audience without a background in pathology or marine mammals, and also don’t translate well to grayscale.

We agree that text describing the tables was duplicative and we have deleted it.

We respectfully disagree with the reviewer regarding the figures. Initially, we had 4 maps to demonstrate the trends in causes of death across the N-S gradient but concluded that there would have been too many figures for this paper. The figures and images will be reformatted in TIF which should improve the image quality. The photographs of the gross findings as well as the blubber sections provide perspective for future efforts to conduct field necropsies and we believe these are essential for the manuscript. The photomicrographs reflect findings of unique microscopic features which to the best of our knowledge, have not been well documented in gray whales.

There is also no attempt to perform any statistical analysis on these data. The authors state that they do not have enough sample points to perform statistical analysis but do not perform any power calculations to back this up. Statistical analysis should be performed and would strengthen the manuscript considerably.

We apologize for making statements that need statistics and have deleted these from the results section.

The lack of statistical analysis is purposeful because this paper is descriptive. If we were comparing age class, sex, or location between UME and non-UME years, statistics would be important. This may be something that could be done in the future when there is sufficient non-UME stranding and PME data has been compiled. Prior to this UME, range wide response to stranded and dead whales was often sporadic and uncoordinated such that the numerators and denominators between this UME and previous years will have to be handled with care. One of the major achievements of this UME investigation has been to coordinate and inform responders from Mexico, the continental US, Canada, and Alaska to support and train each other. 

Coauthors and outside reviewers have asked if the vessel strike animals were more nutritionally compromised than the others, but the sample size really was too small to evaluate nutritional condition (with 4 categories) versus vessel strike in 11 animals. A power analysis would also be confusing here (is the question how many vessel strike whale would we need to prove that more than 50% are thinner than expected on their northbound migration to their foraging grounds? We are still developing case definitions and quantitative analysis protocols to detect and better define thinner than expected whales). 

I think ultimately, as we look at specific high risk areas (San Francisco Bay for example), we can look at the historical data and a growing body of live animal research to understand if vessel strikes have increased. We assume the risk of vessel strike will increase if whales frequent or spend more time in the bay. As a result, the question becomes…does nutritional condition on the wintering grounds predict the likelihood of increased residence time and risk of vessel strike in San Francisco Bay? This would require statistics and modelling and is well beyond the scope of our dataset and the focus of this paper.

I think the most interesting part of this paper is the potential 23 periodicity in UME and unfortunately, this or any other environmental drivers of the UME are not discussed. I understand the division of labor across a diverse research team such as this one and the desire to publish results sooner rather than later. However, I think the manuscript would be strengthened significantly if these mortality results were correlated to the environmental variables in the same manuscript.

As stated previously, the focus of this paper is descriptive pathology of examined whale carcasses and not population modelling or demographics. As mentioned in the comments provided to the other reviewers, a separate paper has recently been published that describes the environmental drivers of the gray whale population numbers. It highlights decreases in the population that correspond with the UMEs and our focus was to characterize the gross and microscopic findings of the stranded animals during the UME. In the prior UME from 1998-2002, to the best of our knowledge, only 3 animals were necropsied, and the intent of our study was to examine as many animals as possible to establish baseline information on health and causes of death during the current event. Without prior data, it is difficult to place our findings in context with respect to whether the incidence of a specific cause of death or potential anthropogenic event is stable, increasing or decreasing over time. This paper reviews post mortem findings of stranded animals provide trends in mortality to compare with ongoing annual, and any future unusual mortality events. Thank you for your comments with regards to the manuscript.

Based on these comments, I am recommending to reject this manuscript and encourage the authors to resubmit their manuscript when the environmental analysis is complete and can be added to these results.

We understand your concern and hope they will be allayed somewhat by the publication of Stewart et al 2023 which models the long term periodicity of the gray whale calf production, mortality, and nutritional condition (based on photogrammetry of live whales) with possible environmental drivers.

We have added the results and citation to the manuscript, but the two datasets cannot be meaningfully combined (multidecadal live animal observations with detailed pathology results from a subset of one mortality event). Efforts are underway to figure out how to augment each other’s datasets, but for now, areas of crossover are elusive (for example, nutritional condition from photogrammetry is always determined looking at the whale from above, while dead whales almost always present with their undersides facing up). Our nutritional condition protocol is an initial attempt to bring these processes together and further blubber analyses (which could be done using live biopsy samples and dead whale blubber) are a next step.

We think that the pathology findings reported in this paper are interesting and important in their own right, especially the changes in blubber pathology with nutritional condition in a fasting animal which is a fascinating topic on its own. This is one of the few papers to ever address the gross and histologic pathology of a large number of stranded baleen whales. That alone is justification for publication, and it will add substantially to the global literature on this topic.

Line 165, sentence reworded to be more concise.

Abstract 

Line 63, spring has been replaced with fall Line 72 and 73, after Two whales, the text has been revised to were entangled at the time of death and one whale died from entrapment. 

Line 75, “could be determined as” and “to have” have been removed.

Line 84, “for the 1999-2000 and current UME” has been removed 

Lines 86-87, “eastern North Pacific gray whale” has been removed. 

Introduction 

line 91 after and, the has been inserted

line 93, after The, latter has been inserted and PCFG whale removed

line 95, after population, have has been replaced with has

line 96, after Russia [1]. The following statement has been inserted. Historically, gray whales were thought to fast during their migration and while resident on the wintering grounds [27]; however, more recent field observations indicate some animals actively forage during the southward and northward migrations [20, 28-30].

Line 103, per another reviewer, greatest has been replaced with largest 

line 107, The 2015/2016 gray whale abundance estimate was 26,960 individuals, which declined to 20,580 in 2019/2020 and further to 16,650 in 2021/2022 [13] has been revised to the text in line 109, The 2015/2016 gray whale abundance estimate was 26,960 individuals, which declined to 20,580 in 2019/2020 and further to 16,650 in 2021/2022 [13].

Line 110, Similar has been replaced with Since, 1967 and later in the sentence, in line 111, from a time series since 1967, occurred has been replaced with “were detected”

Line 113, after 1999-2001 Unusual Mortality Event (UME) has been inserted. Thank you 

Line 114, the paragraph first sentence to has been changed to “Transformative changes in the summer Bering Strait feeding grounds [14], with reported local declines in prey abundance and quality [15], may have a direct impact on gray whale bioenergetics, fecundity, health, and homeostasis [16].”

Line 119, the sentence now starts as Diet varies, by geographic location and whether foraging in shallow mud flats or deeper waters, and consists primarily of benthic and epibenthic invertebrates, including amphipods, crustaceans, mysids, and plankton. References provided, please see response to reviewer 2. 

Line 122, has been replaced with was

Line 125, reference 13 added to the text. For the following sentence, WNP gray whales has been inserted and endangered western gray whale population removed

Line 127, Among the gray whales foraging on the PCFG feeding ground has been removed and a new sentence started with Decreased….after documented, “for PCFG whales” has been inserted. 

Line 131, Eastern North Pacific has been replaced with ENP and gray whale has been removed from before population.

Line 133, the inserted after to and before impacts and environmental change on calf production has been reworded to a changing environment, the editorial suggestion by one reviewer has been reconsidered and not included in the text. 

Line 136, after These, trauma injuries may be, inserted.

Line 139, north changed to North

Line 141 to gray whales has been removed

Lines 144-151, the paragraph has been reworded to the text contained in lines 152-157. The 1999/2001 gray whale UME resulted in 283 confirmed strandings in 1999 and 368 in 2000 [25] however, only three whales were necropsied during this event and while the cause of death for most whales was not determined, malnutrition was considered a prime contributor to the UME [25]. One animal had encephalitis, another had a heavy intestinal helminth infection, and a third had high levels of domoic acid, a biotoxin, in gut contents [30, 31].

Line 157-159, the sentence has been removed and in line 160, the sentence now starts with However, 

Line 161, field observations of has been removed

Line 162 due to the recent increase in abundance has been removed and in line 163, likely has been inserted before contributed. Line 164, dramatic has been removed.

Line 167, As a result… has been replaced with In response to this 

Line 170 an has been changed to the and after UME, the text resulted in a federal investigation that facilitated expert consultation, has been removed and the rest of the sentence continued as facilitated coordination. 

Line 172, after countries, and has been removed

Line 173, Media reports has been removed and replaced as and heighthend public awareness through increased reporting by the media. 

Line 174, the start of the sentence has been changed to Here we present the pathology and ancillary test findings from a …

Lines 175 and 176, and were examined has been removed. 

Materials and Methods

Line 180, the sentence has been reworded to “Marine mammal mortalities observed by the public, biologists, enforcement, and indigenous community members were reported to Canadian, Mexican, or US regional marine mammal stranding response networks, and when feasible, trained personnel were mobilized to examine stranded whales.”

Line 185, were collected has been changed to who collected.

Line 188, after different angles, were used was inserted

Line 191, the sentence has been rephrased to “Interpretation of vessel strike was also based on internal and external examination.” 

Line 193, has been reworded to Age class was initially determined

Line 195 This was replaced the original text, These were

Line 200 has been reworded, after 2-=3 year olds, “therefore final age”, replaced, finally categorized as

Lines 203-212, the text has been reworded for conciseness and the statement “Carcasses were evaluated for nutritional status using a protocol (Appendix 1) established be a review of carcass photographs obtained at the time of necropsy. The new text is

 “Defining emaciation for fasting migratory whales is complicated (for example, blubber depth), while easily measured at necropsy, is not a sensitive measure of nutritional status in gray whales [25]). Because of a lack of available quantitative tests to assess nutritional condition, and to avoid applying different measures of nutrition in different locations, a protocol was developed during this investigation and standardized across all stranding response networks to ensure consistent interpretation of post-mortem nutritional condition (Appendix 1). The protocol evaluated nutritional condition using external features (e.g. the concavity of the epaxial muscle), internal features (e.g. epicardial and mesenteric fat stores), and microscopic evaluation of full thickness blubber sections (e.g. for evidence of adipocyte atrophy or stromal collapse).”

Line 221, the was inserted before nutritional score and the following sentences were modified for clarity to “Carcasses were assigned into one of four nutritional scores: Emaciated (1), thin (2), average (3) and fat (4). Furthermore, the disposition of each stranding at reporting was defined as alive (1), fresh dead (2), moderate decomposition (3), advanced decomposition (4), mummified (5), or undetermined (6) [41].”

To improve the flow of information, the following text was moved to line 226. “Where the carcass code was scored as between decomposition states (e.g., 2.5 or 3.5), [38], the post-mortem state was assigned to the more advanced decomposition state for tabulation and analysis.”

Lines 229-235 have been reworded to “Post-mortem examinations (PMEs) were performed as thoroughly and systematically as possible contingent on multiple variables including, but not limited to, carcass access (eg. beach cast or floating), state of decomposition, human safety (including restrictions imposed by the Covid19 pandemic in 2020), tides, available daylight, and weather. For fresher floating carcasses or those with external evidence of traumatic injuries, efforts were made to tow the animal ashore to a secure and accessible site.”

Line 238, under was replaced with below and later in the sentence, “sampling of” was inserted before skin and blubber. 

Sentences in lines 239-248 were reworded and shortened to follow. “Criteria for inclusion in this study were a full gross PME and, for most cases, sampling for histopathology. Exclusions were based on incomplete PMEs or microscopic assessment of sampled tissues, even if they were confirmed, probable, or suspect cases of human or killer whale interaction. As such, the numbers reported herein are a minimum and represent a subset of the full UME response and investigation.”

Lines 250 -254 revised for conciseness to “Trauma was inferred by evidence of epidermal lacerations or incisions, hemorrhage, edema and necrosis or maceration of skeletal muscle, and skeletal fractures. While entanglement was documented based on presence of foreign material attached to the carcass and determined to have been present at the time of death [43].

Line 256, rephrased and the sentence now reads as. Response level varied and in 

Lines 262-271 reworded to shorten the text to “predation events including missing portions of the tongue or the mandibles, semilunar tissue defects (bite wounds), avulsion of sheets of blubber and skin,”

Line 269, Premortem killer whale was removed, and the sentence reworded to Predation. 

Line 271, after rake marks, “and on histology” was inserted. 

Line 272, The first sentence has been reworded for conciseness to “A case definition of killer whale predation was formulated based on the literature to consistently evaluate the gross lesions and assign a level of diagnostic certainty (Appendix 2).”

Lines 278-282 deleted. 

Line 283 was rephrased to Pigment….. 

Line 284, after re-epithelialization “were used” was inserted 

Line 287 Also considered here was replaced with “Other factors” and after with “the” was inserted. 

Line 290, at the end of the statement “were included in diagnosis of a predation event” was inserted. 

Line 296 the sentence was reworded, and the text shortened and reformatted into two separate statements. “Fixed tissues were trimmed, placed in cassettes and processed using conventional histologic methodology and stained with hematoxylin and eosin (H&E). When indicated, Tort’s Gram stain, Periodic Acid Schiff (PAS) and Perl’s stain (for iron) and reviewed by veterinary pathologists.”

Lines 300-301 to better define the life history samples, the sentence was reworded to “When feasible, baleen, ear plug (wax), stomach contents, full thickness blubber, and ovaries were archived for later life-history studies.”

Lines 315-317, information regarding the use of the open access map for Figure 1 is included and the program details provided. The sentence is “ Reported stranding locations (in decimal degrees) were mapped using ArcGIS Pro 2.8.0 with Natural Earth base maps (naturalearthdata.com).”

Line 333, There were no was inserted before PME data

Lines 335, 349, 378, 387, and 407 the Legend text has been revised with a better explanation of the numbers within the parentheses. 

Line 335: Table 1. Gray whale strandings by location and year (17 December 2018 to 31 December 2021). MX = Mexico, CA=California, OR=Oregon, WA=Washington, BC=British Columbia, AK=Alaska. The number in parentheses is the subset of whales for which a full PME and sampling were conducted. 

Line 353. Table 2. Gray whales stranded by month in Mexico, the United States and Canada. MX = Mexico, CA=California, OR=Oregon, WA=Washington, BC=British Columbia, AK=Alaska. The number in the parentheses is the subset of whales examined and sampled. 

Line 381. Table 3. Gray whales stranded in Mexico, the United States and Canada by age class and geographic location from south to north. MX = Mexico, CA=California, OR=Oregon, WA=Washington, BC=British Columbia, AK=Alaska. The number in the parentheses is the subset of whales included in this study. 

Line 390, Table 4. Gray whale strandings by sex and year. MX = Mexico, CA=California, OR=Oregon, WA=Washington, BC=British Columbia, AK=Alaska. The number in the parentheses is the subset of whales with a full post mortem examination (PME). 

Line 413, Table 5. Gray whales stranded in Mexico, the United States and Canada by decomposition state and geographic location from south to north. MX = Mexico, CA=California, OR=Oregon, WA=Washington, BC=British Columbia, AK=Alaska. Decomposition state is recorded at the time of examination, however for whales that were not examined, this staging refers to the decomposition state at first observation. The number in the parentheses is the subset of whales included in this study. 

Line 366 To reduce repetition, the bulk of the text form the paragraph was removed, and the remainder includes. “The demographics of all the reported gray whales, as well as the subset discussed in this paper (in parentheses) are described in Tables 3 and 4 by location and year. The only newborn noted outside of Mexico was a fetus reported in BC [37].” 

Line 395- to 400, text deleted to minimize repetition. The text now reads “Carcass decomposition varied with 10 mummified animals (2.0%), 251 (50.0%) in advanced decomposition, 175 (34.8%) in moderate decomposition, 51 (10.1%) fresh dead, and 3 (0.6%) that were initially observed alive (Table 5).” 

Line 402, the first statement was moved from the Materials and Methods to Results section. The following statement was revised with the numbers corrected. The text now reads “Of the 61 selected whales, a cause of death was attributed for 33 cases (54%, Table 6, Figs. 1 and 2). This included 16 whales with emaciation as the only post mortem finding, 11 whales with evidence of vessel strike (including two that were also emaciated, three whales with pre-mortem killer whale attack (two probable, one suspect), two entanglement cases, and one entrapment.” 

Line 463, Table 7 inserted at the end of the sentence. 

Line 468, text removed, and the sentence rewritten to could not be determined for various reasons including limited photographic...

Line 477, CBD- Could not be determined added in reference to the last column in the table. 

Line 490-492. The text was reworded to “The reason for the color variation of the blubber was not determined but was observed in the more severely malnourished animals (seven were emaciated and seven were thin).” 

Line 492 was revised to a more active tone to “Stomach contents were not consistently present and/or recorded, but wood chips, bark, eel grass, kelp, and some prey were identified in a few animals” 

Line 495, Gray spelling corrected. 

Line 500, hyphen inserted after plant and before based. 

Line 503 sentence revised and shortened to “The primary cause of death was starvation in 16 of 18 (89%) of the emaciated whales, with no apparent pre-existing pathology that would have contributed to suboptimal nutritional condition.”

Line 509, other inserted after were and before processes and later in the sentence, after in the “death” was inserted. 

Line 514, text removed to shorten sentence to Vessel strike was the cause of death in….

Line 517, after varied for, these cases was inserted.

Line 538, after entrapped in, wharf pilings was inserted 

Line 540, for clarification, that may have resulted was inserted after one entanglement… 

Line 547, after At, necropsy was replaced with PME

Line 549, However removed at the start of the sentence which now reads as Advanced autolysis… 

Line 550, after obscured “similar” was inserted. 

Line 551, revised to have impacted the health…

Lines 558 to 569. To make the text less redundant, a number of phrases have been revised. The new text is 

“Nineteen cases, eight adults and 11 subadults, had lesions consistent with killer whale interaction characterized by acute lacerations or chronic (healed) scars. The latter were observed on 14 whales predominantly on the tips of the fins, flukes, or more rarely along the rostrum. Five carcasses had more acute active lesions consistent with recent killer whale predation: and the likelihood that predation contributed to mortality was categorized as probable for two cases (eg. Figs. 2F and 2G), suspect for one case, and could not be determined in two cases (Fig. 2H).”

Line 570, after for these “five whales were variable.

Line 572, after animals with, chronic non-lethal 

Line 600, to condense the text, the sentence was revised to “Microscopic findings were categorized by organ system and (Table 8) the most common findings. The table number was corrected from 7 to 8. 

Line 608, the sentence was moved to this paragraph. Sporadic and incidental diagnoses included adrenal and myocardial hemorrhage, penile papilloma, myocellular sarcocystosis (Fig. 6D), glossitis and lingual mucosal inclusions (Figs. 6E and 6F), gastric erosions and enteric cestodiasis (Table 8).

Line 628, Table 8, a statement was added to the end of the Legend for clarification “Multiple morphologic diagnoses in individual tissue sections accounted for a total number of diagnoses greater than the 53 of 61 animals that were evaluated microscopically.”

Discussion:

Lines 659-662: For conciseness, the two sentences were changed to 

“The resulting UME declaration by the NMFS enabled this pathologic assessment of stranded whales. While emaciation was speculated to have had a role in the previous 1999/2000 gray whale UME, this investigation confirmed its significant role in the 2019/2021 event. The resulting UME declaration by the NMFS enabled this pathologic assessment of stranded whales.” 

Line 674, to be more specific, achieved through the nutritional condition protocol (SI 1) was inserted.

Lines 678, the sentence was reworded after this investigation, and when combined with validated biomarker measurements should be invaluable with future nutritional health assessments. 

Lines 681-687, the text has been reworded for conciseness. The new paragraph now reads, 

The gradation in atrophy of adipose tissue through the blubber layer reported here suggests that greater effort is required to differentiate atrophy associated with physiologic fasting from atrophy of malnutrition. This would better inform decisions on body condition and enable researchers to factor it into the nutritional condition score matrix in the future (Appendix 1). Because of logistical challenges at the time of PME and varying states of tissue decomposition, not all organs or tissues were systematically examined or collected, so histologic lesions were likely underrepresented. Advanced autolysis in many cases also precluded evaluation of possible endogenous pigments, including endogenous bile or exogenous compounds, such as carotenoids. The accumulation of hepatic iron and atrophy of skeletal muscle in animals stranded in WA and CA substantiated the gross diagnosis of thin or emaciated animals. Hepatocellular hemosiderosis can also be associated with acute infections (with iron sequestration), chronic inflammation, a maladaptive type syndrome, excessive dietary iron consumption, and other factors. However, significant inflammatory lesions were not observed in these cases. 

Line 699, of added before stranded 

Line 700, as well as corrected 

Line 702 revised to more active prose. For example, photogrammetry has detected. And reference 16 added at the end of this statement.

Line 702, after In addition, these methods used was insered. 

Line 705, n=14, one period removed.

line 706, for clarification after northward migration “because they have been fasting” was inserted into the text.

Line 710, a period is inserted after reference 25, and new sentence started with Ectoparasite 

Line 724, per request of reviewer 1 a statement regarding ethical considerations has been inserted. 

Line 790. To convey the information more succinctly, the text related to extralimital detection of gray whales has been reworded and revised. The new paragraph is below. The information has been repositioned and combined with the vessel strike paragraph.

We did not observe an association between suboptimal nutritional status and increased propensity for traumatic injury. Rather, extralimital foraging efforts on the northern migration and increased residence time in coastal bays and sounds on the west coast (e.g. San Francisco Bay, CA, Puget Sound, WA and Port of Los Angeles, CA) increases the time spent in shipping lanes and may increase the risk of vessel strike, exposure to sound disturbances, and possible disorientation. During this UME, gray whales were observed foraging in San Francisco Bay for periods of time up to a month (Markowitz et al. 2022). The increased overlap between whales and shipping lanes is concerning from species conservation and individual animal health and welfare perspectives.

Line 746, text removed, “or the population of gray whales may be approaching carrying capacity. “

Line 748, within replaced with as 

Lines 767-775, to reduce the repetitiveness of the text, the paragraph has been revised and large segments deleted. -

In this case series it is difficult to infer a direct association between the salmon coloration of the blubber observed in WA whales and malnutrition and hepatocellular hemosiderosis. The salmon coloration was distinct from jaundice and was most likely due to exogenous pigment deposition, possibly related to prey shifts and consumption of a diet richer in carotenoids. The subcutaneous edema noted in a few cases of discolored blubber may be secondary to hypoproteinemia, localized trauma, or impaired vascular perfusion as a consequence of live stranding.

Line 802: Further a has been removed and the start of the sentence has been revised to Analysis…the paragraph has also been edited for conciseness and references updated. 

Line 802, Rake marks or scars has been inserted at the start of the sentence and baleen inserted before whale species at the end of the sentence. 

Line 805, this sentence has been with the following text changes: photographic study of free ranging was inserted at the start of the sentence for clarity and with 42% affected added to the end of the sentence. 

Line 806, the start of the sentence has been modified to For fetal predation cases documented in AK, 

Line 809, the sentence has been reordered for clarification, In addition, there may also be and after detachment, or stripping of skin has been removed.

Line 812, and evisceration added at the end of the sentence

Line 815, text removed and the text “bite wounds on the margins of the pectoral flippers and flukes’ inserted.

Line 817, for this and future events was inserted to further substantiate the development of the killer whale case definition

Line 819, will help inserted

Line 822, the text in this sentence was revised for clarity and to limit word usage to 

“External examinations of gray whales that did not meet selection criteria for this study, particularly those stranded in OR where full PME was not always feasible , often revealed substantial evidence of killer whale interactions”

Line 834, after several hours on a beach was revised to before the network was able to respond. 

Line 843, text removed Due to inclement weather, etc, and the sentence revised to Internal examinations can not be pursued

Line 851, in progress replaced with on-going

Line 852, rare inserted before fresh carcasses

Line 855, reference 18 inserted. 

Line 1553, Alaska changed to AK

Line 860, references 57 and 59 inserted

Line 863, reference revised from 40 to 52.

Lines 871-882 and 886 to 896, the text has been revised and shortened to below. 

During the UME, the population of eastern north Pacific gray whales declined from approximately 28,000 animals to 14,000 of which 501 were reported stranded along the western seaboard of Mexico, the United States and Canada. A new analysis modeling the long term periodicity of the gray whale calf production, mortality, and nutritional condition with environmental drivers (Stewart et al, 2023) found that periodic reductions in gray whale numbers were associated with prey availability and access to feeding areas in the Arctic. This model fits with malnutrition as the primary finding in investigations of the 1999-2000 and 2019-2021 events. 

As the Arctic continues to change, the synergy between environmental influences, prey availability, nutritional condition, calf production, and mortality are increasingly important for understanding the fate of the ENP gray whale population. In particular, the impact of oceanic thermal anomalies in the North Pacific [56], unprecedented loss of winter sea ice and extreme summer sea surface temperatures in the Bering and Chukchi marine shelf ecosystems between 2017 and 2019, and changes in infaunal, benthic and pelagic prey abundance and nutritive quality to the morbidity and mortality of gray whales require further research [14-16]. The current data and review of the 2019-2021 strandings, as well as comparison with the epidemiologic and necropsy findings from the 1999-2000 UME, support the hypothesis that these events are likely recurring and transitory. 

Line 897, Finally has been removed and the sentence starts with The roles and citation 57 has been changed to 62

Line 902 The text at the start of the paragraph: “Grey whales continue to strand, and sampling is ongoing with more data and testing to be conducted on archived samples” has been removed. 

Line 905, gray whale population has been inserted and replaces whales

Lines 915-922 have been revised and select text removed for succinctness. 

The preliminary findings from the subset of 61 whales in this study have shown that malnutrition, vessel strikes, killer whale predation, gear entanglement and entrapment were variably implicated. The necropsy findings of malnutrition support the observations of poor body condition in live free ranging whales and low calf recruitment during the UME period suggesting that environmental drivers are impacting the Eastern North Pacific gray whale population.

---

## [Decision Letter · Decision Letter 1]

1 Dec 2023

Gray whale (Eschrichtius robustus) post-mortem findings from December 2018 through 2021 during the Unusual Mortality Event in the Eastern North Pacific

PONE-D-23-22593R1

Dear Dr. Raverty,

We’re pleased to inform you that your manuscript has been judged scientifically suitable for publication and will be formally accepted for publication once it meets all outstanding technical requirements.

Kind regards,

Vitor Hugo Rodrigues Paiva, Ph.D.

Academic Editor

PLOS ONE

Additional Editor Comments (optional):

Reviewers' comments:

Reviewer's Responses to Questions

**Comments to the Author**

1. If the authors have adequately addressed your comments raised in a previous round of review and you feel that this manuscript is now acceptable for publication, you may indicate that here to bypass the “Comments to the Author” section, enter your conflict of interest statement in the “Confidential to Editor” section, and submit your "Accept" recommendation.

Reviewer #2: All comments have been addressed

Reviewer #3: All comments have been addressed

2. Is the manuscript technically sound, and do the data support the conclusions?

Reviewer #2: Yes

Reviewer #3: Yes

3. Has the statistical analysis been performed appropriately and rigorously? 

Reviewer #2: Yes

Reviewer #3: Yes

4. Have the authors made all data underlying the findings in their manuscript fully available?

Reviewer #2: Yes

Reviewer #3: Yes

5. Is the manuscript presented in an intelligible fashion and written in standard English?

Reviewer #2: Yes

Reviewer #3: Yes

6. Review Comments to the Author

Reviewer #2: The manuscript has improved considerably. I answered yes to the quesition concerning statistical analysis. There are no hypoteses to be tested, but the conclusions are based on sound data.

Reviewer #3: I thoroughly read your manuscript and think that you have improved it significantly. The revisions you've made have strengthened the overall structure and coherence of the content, making it more engaging and compelling. I recommend this manuscript for publication.

7. PLOS authors have the option to publish the peer review history of their article (what does this mean?). If published, this will include your full peer review and any attached files.

Reviewer #2: No

Reviewer #3: No

---

## [Editor Report · Acceptance letter]

26 Feb 2024

PONE-D-23-22593R1 

PLOS ONE

Dear Dr. Raverty, 

I'm pleased to inform you that your manuscript has been deemed suitable for publication in PLOS ONE. Congratulations! Your manuscript is now being handed over to our production team.

Kind regards, 

on behalf of

Dr. Vitor Hugo Rodrigues Paiva 

Academic Editor

PLOS ONE